# Distance Encoding: Design Provably More Powerful Neural Networks for Graph Representation Learning

**Pan Li**
Department of Computer Science
Purdue University
panli@purdue.edu

**Yanbang Wang**
Department of Computer Science
Stanford University
ywangdr@cs.stanford.edu

**Hongwei Wang**
Department of Computer Science
Stanford University
hongweiw@cs.stanford.edu

**Jure Leskovec**
Department of Computer Science
Stanford University
jure@cs.stanford.edu

## Abstract

Learning representations of sets of nodes in a graph is crucial for applications ranging from node-role discovery to link prediction and molecule classification. Graph Neural Networks (GNNs) have achieved great success in graph representation learning. However, expressive power of GNNs is limited by the 1-Weisfeiler-Lehman (WL) test and thus GNNs generate identical representations for graph substructures that may in fact be very different. More powerful GNNs, proposed recently by mimicking higher-order-WL tests, only focus on representing entire graphs and they are computationally inefficient as they cannot utilize sparsity of the underlying graph. Here we propose and mathematically analyze a general class of structure-related features, termed Distance Encoding (DE). DE assists GNNs in representing any set of nodes, while providing strictly more expressive power than the 1-WL test. DE captures the distance between the node set whose representation is to be learned and each node in the graph. To capture the distance DE can apply various graph-distance measures such as shortest path distance or generalized PageRank scores. We propose two ways for GNNs to use DEs (1) as extra node features, and (2) as controllers of message aggregation in GNNs. Both approaches can utilize the sparse structure of the underlying graph, which leads to computational efficiency and scalability. We also prove that DE can distinguish node sets embedded in almost all regular graphs where traditional GNNs always fail. We evaluate DE on three tasks over six real networks: structural role prediction, link prediction, and triangle prediction. Results show that our models outperform GNNs without DE by up-to 15% in accuracy and AUROC. Furthermore, our models also significantly outperform other state-of-the-art methods especially designed for the above tasks.

## 1 Introduction

Graph representation learning aims to learn representation vectors of graph-structured data [1]. Representations of node sets in a graph can be leveraged for a wide range of applications, such as discovery of functions/roles of nodes based on individual node representations [2–6], link or link type prediction based on node-pair representations [7–10] and graph comparison or molecule classification based on entire-graph representations [11–17].

Graph neural networks (GNNs), inheriting the power of neural networks [18], have become the *de facto* standard for representation learning in graphs [19]. Generaly, GNNs use message passing procedure over the input graph, which can be summarized in three steps: (1) Initialize node representations with their initial attributes (if given) or structural features such as node degrees;

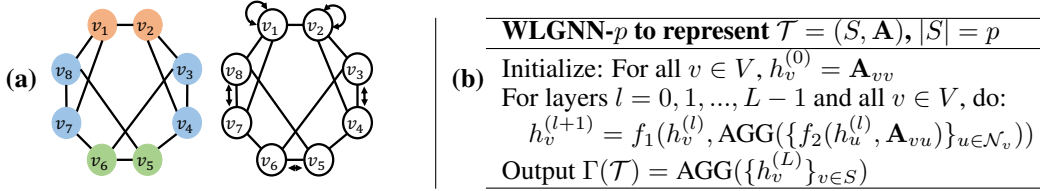

Figure 1: **(a)** 3-regular graph with 8 nodes. Briefly assume that all node attributes are the same and the nodes can only be distinguished based on their network structure. Then for all nodes, WLGNN will produce the same representation and thus fail to distinguish them. However, nodes with different colors should have different representations, as they are not structurally equivalent (or "isomorphic" as defined in Section 2). Furthermore, WLGNN cannot distinguish all the node-pairs (*e.g.*, $\{v_1, v_2\}$ vs $\{v_4, v_7\}$). However, if we use shortest-path-distances (SPDs) between nodes as features we can distinguish blue nodes from green and red nodes because there is another node with SPD$= 3$ to a blue node of interest (*e.g.*, SPD between $v_3$ and $v_8$), while all SPDs between other nodes to red/green nodes are less than 3. Note that the structural equivalence between any two nodes of the same color can be obtained from the reflexivity of the graph while the equivalence between two vertically-aligned blue nodes can be further obtained from the node permutation shown in the right. **(b)** WLGNN algorithm to represent a node set $S$ of size $p$ — $f_i(\cdot)$'s are arbitrary neural networks; AGG$(\cdot)$'s are set-pooling operators; $L$ is the number of layers.

(2) Iteratively update the representation of each node by aggregating over the representations of its neighboring nodes; (3) Readout the final representation of a single node, a set of nodes, or the entire node set as required by the task. Under the above framework, researchers have proposed many GNN architectures [14–16, 20–23]. Interested readers may refer to tutorials on GNNs for further details [1, 24].

Despite the success of GNNs, their representation power in representation learning is limited [16]. Recent works proved that the representation power of GNNs that follow the above framework is bounded by the 1-WL test [16, 25, 26] (We shall refer to these GNNs as WLGNNs). Concretely, WLGNNs yield identical vector representations for any subgraph structure that the 1-WL test cannot distinguish. Consider an extreme case: If node attributes are all nodes are the same, then for any node in a $r$-regular graph GNN will output identical representation. Such an issue becomes even worse when WLGNNs are used to extract representations of node sets, *e.g.*, node-pairs for link prediction (Fig. 1 (a)). A few works have been recently proposed to improve the power of WLGNNs [27]. However, they either focus on building theory only for entire-graph representations [26–30], or show empirical success using heuristic methods without strong theoretical characterization [9,10,17,31–33]. We review these methods in detail in Section 4.

Here we address the limitations of WLGNNs and propose and mathematically analyze a new class of node features, termed *Distance Encoding (DE)*. DE comes with both theoretical guarantees and empirical efficiency. Given a node set $S$ whose structural representation is to be learnt, for every node $u$ in the graph DE is defined as a mapping of a set of landing probabilities of random walks from each node of the set $S$ to node $u$. DE may use measures such as shortest path distance (SPD) and generalized PageRank scores [34]. DE can be combined with any GNN architecture in simple but effective ways: First, we propose DE-GNN that utilizes DE as an extra node feature. We further enhance DE-GNN by allowing DE to control the message aggregation procedure of WLGNNs, which yields another model DEA-GNN. Since DE purely depends on the graph structure and is independent of node identifiers, DE also provides inductive and generalization capability.

We mathematically analyze the expressive power of DE-GNN and DEA-GNN for structural representation learning. We prove that the two models are able to distinguish two non-isomorphic equally-sized node sets (including nodes, node-pairs, . . . , entire-graphs) that are embedded in almost all sparse regular graphs, where WLGNN always fails to distinguish them unless discriminatory node/edge attributes are available. We also prove that the two models are not more powerful than WLGNN when applied to distance regular graphs [35], which implies the limitation of DEs. However, we show that DE has an extra power to learn the structural representations of node-pairs over distance regular graphs [35].

We experimentally evaluate DE-GNN and DEA-GNN on three levels of tasks including node-structural-role classification (node-level), link prediction (node-pair-level), triangle prediction (node-triad-level). Our methods outperform WLGNN on all three tasks by up-to 15% improvement in average accuracy. Our methods also outperform other baselines specifically designed for these tasks.

## 2 Preliminaries

In this section we formally define the notion of structural representation and review how WLGNN learns structural representation and its relation to the 1-WL test.

### 2.1 Graph Representation Learning

**Definition 2.1.** We consider an undirected graph which can be represented as $G = (V, E, \mathbf{A})$, where $V = [n]$ is the node set, $E \subseteq V \times V$ is the edge set, and $\mathbf{A}$ contains all features in the space $\mathcal{A} \subset \mathbb{R}^{n \times n \times k}$. Its diagonal component, $\mathbf{A}_{vv.}$, denotes the node attributes of node $v(\in V)$, while its off-diagonal component in $\mathbf{A}_{vu.}$ denotes the node-pair attributes of $(v, u)$. We set $\mathbf{A}_{vu.}$ as all zeros if $(v, u) \notin E$. In practice, graphs are usually sparse, *i.e.*, $|E| \ll n^2$. We introduce $A \in \{0, 1\}^{n \times n}$ to denote the adjacency matrix of $G$ such that $A_{uv} = 1$ iff $(u, v) \in E$. Note that $A$ can be also viewed as one slice of the feature tensor $\mathbf{A}$. If no node/edge attributes are available, we let $\mathbf{A} = A$.

**Definition 2.2.** The node permutation denoted by $\pi$ is a bijective mapping from $V$ to $V$. All possible $\pi$'s are collected in the permutation group $\Pi_n$. We denote $\pi$ acting on a subset $S$ of $V$ as $\pi(S) = \{\pi(i) | i \in S\}$. We further define $\pi(\mathbf{A})_{uv.} = \mathbf{A}_{\pi^{-1}(u)\pi^{-1}(v).}$ for any $(u, v) \in V \times V$.

**Definition 2.3.** Denote all $p$-sized subsets $S$ of $V$ as $S \in \mathcal{P}_p(V)$ and define the space $\Omega_p = \mathcal{P}_p(V) \times \mathcal{A}$. For two tuples $\mathcal{T}_1 = (S^{(1)}, \mathbf{A}^{(1)})$ and $\mathcal{T}_2 = (S^{(2)}, \mathbf{A}^{(2)})$ in $\Omega_p$, we call that that they are *isomorphic* (otherwise *non-isomorphic*), if $\exists \pi \in \Pi_n$ such that $S^{(1)} = \pi(S^{(2)})$ and $\mathbf{A}^{(1)} = \pi(\mathbf{A}^{(2)})$.

**Definition 2.4.** A function $f$ defined on $\Omega_p$ is *invariant* if $\forall \pi \in \Pi_n, f(S, \mathbf{A}) = f(\pi(S), \pi(\mathbf{A}))$.

**Definition 2.5.** The *structural representation* of *a tuple* $(S, \mathbf{A})$ is an invariant function $\Gamma(\cdot) : \Omega_p \to \mathbb{R}^d$ where $d$ is the dimension of representation. Therefore, if two tuples are isomorphic, they should have the same structural representation.

The invariant property is critical for the inductive and generalization capability as it frees structural representations from node identifiers and effectively reduces the problem dimension by incorporating the symmetry of the parameter space [29] (*e.g.*, the convolutional layers in GCN [20]). The invariant property also implies that structural representations do not allow encoding the absolute positions of $S$ in the graph.

The definition of structural representation is very general. Suppose we set two node sets $S^{(1)}$, $S^{(2)}$ as two single nodes and set two graph structures $\mathbf{A}^{(1)}$ and $\mathbf{A}^{(2)}$ as the ego-networks around these two nodes. Then, the definition of structural representation provides a mathematical characterization the concept "structural roles" of nodes [3, 5, 6], where two far-away nodes could have the same structural roles (representations) as long as their ego-networks have the same structure.

Note that depending on the application one can vary/select the size $p$ of the node set $S$. For example, when $p = 1$ then we are in the regime of node classification, $p = 2$ is link prediction, and when $S = V$, structural representations reduce to entire graph representations. However, in this work we will primarily focus on the case that the node set $S$ has a fixed and small size $p$, where $p$ does not depend on the graph size $n$. Although Corollary 3.4 later shows the potential of our techniques on learning the entire graph representations, this is not the main focus of our work here. We expect the techniques proposed here can be further used for entire-graph representations while we leave the detailed investigation for future work.

Although structural representation defines a more general concept, it shares some properties with traditional entire-graph representation. For example, the universal approximation theorem regarding entire-graph representation [30] can be directly generalized to the case of structural representations:

**Theorem 2.6.** If structural representations $\Gamma$ are different over any two non-isomorphic tuples $\mathcal{T}_1$ and $\mathcal{T}_2$ in $\Omega_p$, then for any invariant function $f : \Omega_p \to \mathbb{R}$, $f$ can be universally approximated by feeding $\Gamma$ into a 3-layer feed-forward neural network with ReLu as the activation function, as long as (1) the feature space $\mathcal{A}$ is compact and (2) $f(S, \cdot)$ is continuous over $\mathcal{A}$ for any $S \in \mathcal{P}_p(V)$.

Theorem 2.6 formally establishes the relation between learning structural representations and distinguishing non-isomorphic structures, *i.e.*, $\Gamma(\mathcal{T}_1) \neq \Gamma(\mathcal{T}_2)$ iff $\mathcal{T}_1$ and $\mathcal{T}_2$ are non-isomorphic. However, no polynomial algorithm has been found to distinguish even just non-isomorphic entire graphs ($S = V$) without node/edge attributes ($\mathbf{A} = A$), which is known as the graph isomorphism problem [36]. In this work, we will use the range of non-isomorphic structures that GNNs can distinguish to characterize their expressive power for graph representation learning.

## 2.2 Weisfeiler-Lehman Tests and WLGNN for Structural Representation Learning

Weisfeiler-Lehman test (WL-test) is a family of very successful algorithmic heuristics used in graph isomorphism problems [25]. 1-WL test, the simplest one among this family, starts with coloring nodes with their degrees, then it iteratively aggregates the colors of nodes and their neighborhoods, and hashes the aggregated colors into unique new colors. The coloring procedure finally converges to some static node-color configuration. Here a node-color configuration is a multiset that records the types of colors and their numbers. Different node-color configurations indicate two graphs are non-isomorphic while the reverse statement is not always true.

More than the graph isomorphism problem, the node colors obtained by the 1-WL test naturally provide a test of structural isomorphism. Consider two tuples $\mathcal{T}_1 = (S^{(1)}, \mathbf{A}^{(1)})$ and $\mathcal{T}_2 = (S^{(2)}, \mathbf{A}^{(2)})$ according to Definition 2.3. We temporarily ignore node/edge attributes for simplicity, so $\mathbf{A}^{(1)}, \mathbf{A}^{(2)}$ reduce to adjacent matrices. It is easy to show that different node-color configurations of nodes in $S^{(1)}$ and in $S^{(2)}$ obtained by the 1-WL test also indicate that $\mathcal{T}_1$ and $\mathcal{T}_2$ are not isomorphic.

WLGNNs refer to those GNNs that mimic the 1-WL test to learn structural representation, which is summarized in Fig. 1 (b). It covers many well-known GNNs of which difference may appear in the implementation of neural networks $f_i$ and set-poolings AGG($\cdot$) (Fig. 1 (b)), including GCN [20], GraphSAGE [21], GAT [22], MPNN [14], GIN [16] and many others [29]. Note that we use WLGNN-$p$ to denote the WLGNN that is to learn structural representations of node sets $S$ with size $|S| = p$. One may directly choose $S = V$ to obtain the entire-graph representation. Theoretically, the structural representation power of WLGNN-$p$ is provably bounded by the 1-WL test [16]. The result can be also generalized to the case of structural representations as follows.

**Theorem 2.7.** Consider two tuples $\mathcal{T}_1 = (S^{(1)}, \mathbf{A}^{(1)})$ and $\mathcal{T}_2 = (S^{(2)}, \mathbf{A}^{(2)})$ in $\Omega_p$. If $\mathcal{T}_1, \mathcal{T}_2$ cannot be distinguished by the 1-WL test, then the corresponding outputs of WLGNN-$p$ satisfy $\Gamma(\mathcal{T}_1) = \Gamma(\mathcal{T}_2)$. On the other side, if they can be distinguished by the 1-WL test and we suppose aggregation operations (AGG) and neural networks $f_1$, $f_2$ are all injective mappings, then with a large enough number of layers $L$, the outputs of WLGNN-$p$ also satisfy $\Gamma(\mathcal{T}_1) \neq \Gamma(\mathcal{T}_2)$.

Because of Theorem 2.7, WLGNN inherits the limitation of the 1-WL test. For example, WLGNN cannot distinguish two equal-sized node sets in all $r$-regular graphs (unless node/edge features are discriminatory). Here, a $r$-regular graph means that all its nodes have degree $r$. Therefore, researchers have recently focused on designing GNNs with expressive power greater than the 1-WL test. Here we will improve the power of GNNs by developing a general class of structural features.

# 3 Distance Encoding and Its Power

## 3.1 Distance Encoding

Suppose we aim to *learn the structural representation of the target node set $S$*. Intuitively, our proposed DE will then encode the distance from $S$ to any other node $u$. We define DE as follows:

**Definition 3.1.** Given a target set of nodes $S \in 2^V \backslash \emptyset$ of $G$ with the adjacency matrix $A$, we denote *distance encoding* as a function $\zeta(\cdot|S, A) : V \to \mathbb{R}^k$. $\zeta$ should also be permutation invariant, *i.e.*, $\zeta(u|S, A) = \zeta(\pi(u)|\pi(S), \pi(A))$ for all $u \in V$ and $\pi \in \Pi_n$. Then we denote DEs *w.r.t.* the size of $S$ and call them as *DE-p* if $|S| = p$.

Later we use $\zeta(u|S)$ for brevity where $A$ could be inferred from the context. For simplicity, we choose DE as a set aggregation (*e.g.*, the sum-pooling) of DEs between nodes $u$, $v$ where $v \in S$:

$$\zeta(u|S) = \text{AGG}(\{\zeta(u|v)|v \in S\}) \tag{1}$$

More complicated DE may be used while this simple design can be efficiently implemented and achieves good empirical performance. Then, the problem reduces to choosing a proper $\zeta(u|v)$. Again for simplicity, we consider the following class of functions that is based on the mapping of a list of landing probabilities of random walks from $v$ to $u$ over the graph, *i.e.*,

$$\zeta(u|v) = f_3(\ell_{uv}), \; \ell_{uv} = ((W)_{uv}, (W^2)_{uv}, ..., (W^k)_{uv}, ...) \tag{2}$$

where $W = AD^{-1}$ is the random walk matrix, $f_3$ may be simply designed by some heuristics or be parameterized and learnt as a feed-forward neural network. In practice, a finite length of $\ell_{vu}$, say 3,4, is enough. Note that Eq. (2) covers many important distance measures. First, setting $f_3(\ell_{uv})$ as

the first non-zero position in $\ell_{uv}$ gives the *shortest-path-distance (SPD)* from $v$ to $u$. We denote this specific choice as $\zeta_{spd}(u|v)$. Second, one may also use *generalized PageRank scores* [34]:

$$\zeta_{gpr}(u|v) = \sum_{k \geq 1} \gamma_k (W^k)_{uv} = (\sum_{k \geq 1} \gamma_k W^k)_{uv}, \quad \gamma_k \in \mathbb{R}, \text{ for all } k \in \mathbb{N} . \qquad (3)$$

Note that the permutation invariant property of DE is beneficial for inductive learning, which fundamentally differs from positional node embeddings such as node2vec [37] or one-hot node identifiers. In the rest of this work, we will show that DE improves the expressive power of GNNs in both theory and practice. In Section 3.2, we use DE as extra node features. We term this model as DE-GNN, and theoretically demonstrate its expressive power. In the next subsection, we further use DE-1 to control the aggregation procedure of WLGNN. We term this model as DEA-GNN and extend our theory there.

## 3.2 DE-GNN— Distance Encodings as Node Features

DE can be used as extra node features. Specifically, we improve WLGNNs by setting $h_v^{(0)} = \mathbf{A}_{vv} \oplus \zeta(v|S)$ where $\oplus$ is the concatenation. We call the obtained model DE-GNN. We similarly use **DE-GNN-$p$** to specify the case when $|S| = p$. For simplicity, we give the following definition.

**Definition 3.2.** DE-GNN is called *proper* if $f_1, f_2$, AGGs in the WLGNN (Fig. 1 (b)), and AGG in Eq. (1), $f_3$ in Eq. (2) are injective mappings as long as the input features are all countable.

We know that a proper DE-GNN exists because of the universal approximation theorem of feed-forward networks (to construct $f_i$, $i \in \{1, 2, 3\}$) and Deep Sets [38] (to construct AGGs).

### 3.2.1 The Expressive Power of DE-GNN

Next, we demonstrate the power of DE-GNN to distinguish structural representations. Recall that the fundamental limit of WLGNN is the 1-WL test for structural representation (Theorem 2.7). One important class of graphs that cannot be distinguished by the 1-WL test are regular graphs (although, in practice, node/edge attributes may help diminish such difficulty by breaking the symmetry). In theory, we may consider the most difficult case by assuming that no node/edge attributes are available. In the following, our main theorem shows that even in the most difficult case, DE-GNN is able to distinguish two equal-sized node sets that are embedded in almost all $r$-regular graphs. One example where DE-GNN using $\zeta_{spd}(\cdot)$ (SPD) is shown in Fig. 1 (a): The blue nodes can be easily distinguished from the green or red nodes as SPD= 3 may appear between two nodes when a blue node is the node set of interest, while all SPDs from other nodes to red and green nodes are less than 3. Actually, DE-GNN-1 with SPD may also distinguish the red or green nodes by investigating its procedure in details (Fig. 3 in Appendix).

**Theorem 3.3.** Given two equal-sized sets $S^{(1)}, S^{(2)} \subset V$, $|S^{(1)}| = |S^{(2)}| = p$. Consider two tuples $\mathcal{T}^{(1)} = (S^{(1)}, \mathbf{A}^{(1)})$ and $\mathcal{T}^{(2)} = (S^{(2)}, \mathbf{A}^{(2)})$ in the most difficult setting where features $\mathbf{A}^{(1)}$ and $\mathbf{A}^{(2)}$ are only different in graph structures specified by $A^{(1)}$ and $A^{(2)}$ respectively. Suppose $A^{(1)}$ and $A^{(2)}$ are uniformly independently sampled from all r-regular graphs over $V$ where $3 \leq r < (2 \log n)^{1/2}$. Then, for any small constant $\epsilon > 0$, within $L \leq \lceil (\frac{1}{2} + \epsilon) \frac{\log n}{\log(r-1)} \rceil$ layers, there exists a proper DE-GNN-$p$ using DEs $\zeta(u|S^{(1)})$, $\zeta(u|S^{(2)})$ for all $u \in V$, such that with probability $1 - o(n^{-1})$, the outputs $\Gamma(\mathcal{T}^{(1)}) \neq \Gamma(\mathcal{T}^{(2)})$. Specifically, $f_3$ can be simply chosen as SPD, *i.e.*, $\zeta(u|v) = \zeta_{spd}(u|v)$. The big-O notations here and later are w.r.t. $n$.

*Remark* 3.1. In some cases, we are to learn representations of structures that lie in a single large graph, *i.e.*, $A^{(1)} = A^{(2)}$. Actually, there is no additional difficulty to extend the proof of Theorem 3.3 to this setting as long as $A^{(1)}(= A^{(2)})$ is uniformly sampled from all r-regular graphs and $S^{(1)} \cap S^{(2)} = \emptyset$. The underlying intuition is that for large $n$, the local subgraphs (within $L$-hop neighbors) around two non-overlapping fixed sets $S^{(1)}, S^{(2)}$ are almost independent. Simulation results to validate the single node case ($p = 1$) of Theorem 3.3 and Remark 3.1 are shown in Fig. 2 (a).

Actually, the power of structural representations of small node sets can be used to further characterize the power of entire graph representations. Consider that we directly aggregate all the representations of nodes of a graph output by DE-GNN-1 via set-pooling as the graph representation, which is a common strategy adopted to learn graph representation via WLGNN-1 [14, 16, 23]. So how about the power of DE-GNN-1 to represent graphs? To answer this question, suppose two $n$-sized $r$-regular graphs $\mathbf{A}^{(1)}$

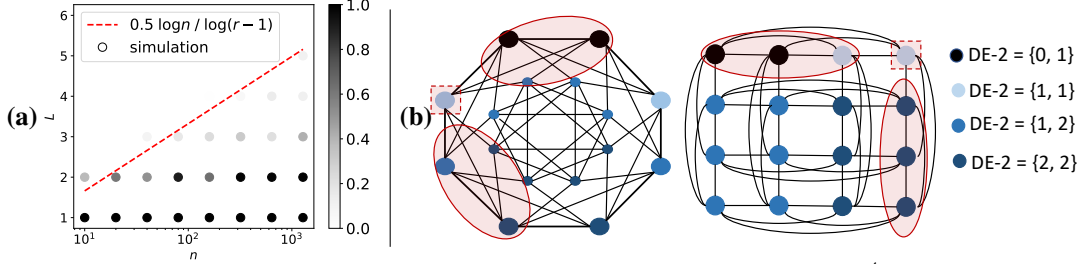

Figure 2: **(a)** Simulation to validate Theorem 3.3. We uniformly at random generate $10^4/n$ 3-regular graphs and compare node representations output by a randomly initialized but untrained DE-GNN-1 with $L$ layers, $L \leq 6$. All the nodes in these graphs are considered and thus for each $n$, there are $10^4$ nodes from the same or different graphs. For any two nodes $u, v$, if $\|h_u^{(L)} - h_v^{(L)}\|_2$ is greater than machine accuracy, they are regarded to be distinguishable. The colors of the scatter plot indicate the portion of two nodes that are not distinguishable by DE-GNN-1. The red line is boundary predicted by our theory, which well matches the simulation. **(b)** The power of DE-2. The left is the Shrikhande graph while the right is the $4 \times 4$ Rook's graph. DE-GNN-1 assigns all nodes with the same representation. DE-GNN-2 may distinguish the structures by learning representations of node-pairs (edges)—the node-pairs colored by black. Each node is colored with its DE-2 that is a set of SPDs to either node in the target edges (Eq. (1)). Note the neighbors of nodes with DE-2= $\{1, 1\}$ (covered by dashed boxes) that are highlighted by red ellipses. As these neighbors have different DE-2's, after one layer of DE-GNN-2, the intermediate representations of nodes with DE-2= $\{1, 1\}$ are different between these two graphs. Using another layer, DE-GNN-2 can distinguish the representations of two target edges.

and $\mathbf{A}^{(2)}$ satisfy the condition in Theorem 3.3. Then, by using a union bound, Theorem 3.3 indicates that for a node $v \in V$, its representation $\Gamma((v, \mathbf{A}^{(1)})) \notin \{\Gamma((u, \mathbf{A}^{(2)})) | u \in V\}$ with probability $1 - no(n^{-1}) = 1 - o(1)$. Therefore, these two graphs $\mathbf{A}^{(1)}$ and $\mathbf{A}^{(2)}$ can be distinguished via DE-GNN-1 with high probability. We formally state this result in the following corollary.

**Corollary 3.4.** Suppose two graphs are uniformly independently sampled from all $n$-sized $r$-regular graphs over $V$ where $3 \leq r < (2 \log n)^{1/2}$. Then, within $L \leq \lceil (\frac{1}{2} + \epsilon) \frac{\log n}{\log(r-1)} \rceil$ layers, DE-GNN-1 can distinguish these two graphs with probability $1 - o(1)$ by being concatenated with injective set-pooling over all the representations of nodes.

One insightful observation is that the structural representation with a small set $S$ may become easier to be learnt than the entire graph representation as the knowledge of the node set $S$ can be viewed as a piece of side information. Models can be built to leverage such information, while when to compute the entire graph representation ($S = V$), the model loses such side information. This could be a reason why the successful probability to learn structural representation of a small node set (Theorem 3.3) is higher than to that of an entire graph in (Corollary 3.4), though our derivation of the probabilistic bounds is not tight. Note that DE provides a convenient way to effectively capture such side information. One naive way to leverage the information of $S$ is to simply annotate the nodes in $S$ with binary encoding 1 and those out of $S$ with 0. This is obviously a special case of DE but it is not as powerful as the general DE (even compared to the special case SPD). Think about setting the target node set in Eq.(1) as the entire graph ($S = V$), where annotating the nodes in/out of $S$ does not improve the representation power of WLGNN because all nodes are annotated as 1. However, DE still holds extra representation power: For example, we want to distinguish a graph with two disconnected 3-circles and a graph with a 6-circle. These two graphs generate different SPDs between nodes.

### 3.2.2 The Limitation of DE-GNN

Next, we show the limitation of DE-GNN. We prove that over a subclass of regular graphs, distance regular graphs (DRG), DE-1 is useless for structural representation. We provide the definition of DRG as follows while we refer interested readers to check more properties of DRGs in [35].

**Definition 3.5.** A *distance regular graph* is a regular graph such that for any two nodes $v, u \in V$, the number of vertices $w$ s.t. $\text{SPD}(w, v) = i$ and $\text{SPD}(w, u) = j$, only depends on $i, j$ and $\text{SPD}(v, u)$.

The Shrikhande graph and the $4 \times 4$ Rook's graph are two non-isomorphic DRGs shown in Fig. 2 (b) (We temporarily ignore the nodes colors which will be discussed later). For simplicity, we only consider connected DRGs that can be characterized by arrays of integers termed intersection arrays.

**Definition 3.6.** The *intersection array* of a connected DRG with diameter $\triangle$ is an array of integers $\{b_0, b_1, ..., b_{\triangle-1}; c_1, c_2, ..., c_\triangle\}$ such that for any node pair $(u, v) \in V \times V$ that satisfies $\text{SPD}(v, u) = j$, $b_j$ is the number of nodes $w$ that are neighbors of $v$ and satisfy $\text{SPD}(w, u) = j + 1$, and $c_j$ is the number of nodes $w$ that are neighbors of $v$ and satisfy $\text{SPD}(w, u) = j - 1$.

It is not hard to show that the two DRGs in Fig. 2 (b) share the same intersecion array $\{6, 3; 1, 2\}$. The following theorem shows that over distance regular graphs, DE-GNN-1 requires discriminatory node/edge attributes to distinguish structures, which indicates the limitation of DE-1.

**Theorem 3.7.** Given any two nodes $v, u \in V$, consider two tuples $\mathcal{T}_1 = (v, \mathbf{A}^{(1)})$ and $\mathcal{T}_2 = (u, \mathbf{A}^{(2)})$ with graph structures $A^{(1)}$ and $A^{(2)}$ that correspond to two connected DRGs with a same intersection array. Then, DE-GNN-1 must use discriminatory node/edge attributes to distinguish $\mathcal{T}_1$ and $\mathcal{T}_2$.

Note Theorem 3.7 only works for node representations using DE-1. Therefore, DE-GNN-1 may not associate distinguishable node representations in the two DRGs in Fig. 2 (b).

However, if we are to learn higher-order structural representations ($|S| \geq 2$) with DE-$p$ ($p \geq 2$), DE-GNN-p may have even stronger representation power. We illustrate this point by considering structural representations of two node-pairs that form edges of the two DRGs respectively. Consider two node-pairs that correspond to two edges of these two graphs in Fig. 2 (b) respectively. Then, there exists a proper DE-GNN-2 via using SPD as DE-2, associating these two node-pairs with different representations. Moreover, by simply aggregating the obtained representations of all node-pairs into graph representations via a set-pooling, we may also distinguish these two graphs. Note that distinguishing the node-pairs of the two DRGs is really hard, because even the 2-WL test [1] will fail to distinguish any edges in the DRGs with a same intersection array and diameters exactly equal to 2 [2]. This means that the recently proposed more powerful GNNs, such as RingGNN [30] and PPGN [27], will also fail in this case. However, it is possible to use DE-GNN-2 to distinguish those two DRGs.

It is interesting to generalize Theorem 3.3 to DRGs to demonstrate the power of DE-GNN-$p$ ($p \geq 2$). However, missing analytic-friendly random models for DRGs makes such generalization challenging.

### 3.3   DEA-GNN— Distance Encoding-1's as Controllers of the Message Aggregation

DE-GNN only uses DEs as initial node features. In this subsection, we further consider leveraging DE-1 between any two nodes to control the aggregation procedure of DE-GNN. Specifically, we propose DE-Aggregation-GNN (DEA-GNN) to do the following change

$$\text{AGG}(\{f_2(h_u^{(l)}, \mathbf{A}_{vu})\}_{u \in \mathcal{N}_v}) \rightarrow \text{AGG}(\{(f_2(h_u^{(l)}, \mathbf{A}_{vu}), \zeta(u|v))\}_{u \in V}) \tag{4}$$

Note that the representation power of DEA-GNN is at least no worse than DE-GNN because the later one is specialized by aggregating the nodes with $\zeta_{spd}(u|v) = 1$, so Theorem 3.3, Corollary 3.4 are still true. Interestingly, its power is also limited by Theorem 3.7. We conclude as the follows.

**Corollary 3.8.** Theorem 3.3, Corollary 3.4 and Theorem 3.7 are still true for DEA-GNN.

The general form Eq. (4) that aggregates all nodes in each iteration holds more theoretical significance than practical usage due to scalability concern. In practice, the aggregation procedure of DEA-GNN may be trimmed by balancing the tradeoff between complexity and performance. For example, we may choose $\zeta(u|v) = \zeta_{spd}(u|v)$, and only aggregate the nodes $u$ such that $\zeta_{spd}(u|v) \leq K$, *i.e.*, $K$-hop neighbors. Multi-hop aggregation allows avoiding the training issues of deep architecture, *e.g.*, gradient degeneration. Particularly, we may prove that $K$-hop aggregation decreases the number of layers $L$ requested to $\lceil (\frac{1}{2} + \epsilon) \frac{\log n}{K \log(r-1)} \rceil$ in Theorem 3.3 and Corollary 3.4 with proof in Appendix F. We may also choose $\zeta(u|v) = \zeta_{gpr}(u|v)$ with non-negative $\gamma_k$ in Eq. (3) and aggregate the nodes whose $\zeta(u|v)$ are top-$K$ ranked among all $u \in V$. This manner is able to control fix-sized aggregation sets. As DEA-GNN does not show provably better representation power than DE-GNN, all the above approaches share the same theoretical power and limitations. However, in practice their specific performance may vary across datasets and applications.

## 4 Related Work

Recently, extensive effort has been taken to improve the structural representation power of WL-GNN. From the theoretical perspective, most previous works only considered representations of entire graphs [26–30, 42, 43] while Srinivasan & Ribeiro initialized the investigation of structural representations of node sets [44] from the view of joint probabilistic distributions. Some works view GNNs as general approximators of invariant functions but the proposed models hold more theoretical implication than practical usage because of their dependence on polynomial$(n)$-order tensors [29, 45, 46]. Ring-GNN [30] (or equivalently PPGN [27]), a relatively scalable model among them, was based on 3-order tensors and was proposed to achieve the expressive power of the 2-WL test (a brief introduction in Appendix H). However, Ring-GNN (PPGN) was proposed for entire-graph representations and cannot leverage the sparsity of the graph structure to be scalable enough to process large graphs [27, 30]. DE-GNN still benefits from such sparsity and are also used to represent node sets of arbitrary sizes. Moreover, our models theoretically behave orthogonal to Ring-GNN, as DE-2 can distinguish some non-isomorphic node-pairs that Ring-GNN fails to distinguish because the power of Ring-GNN is limited by the 2-WL test (Fig. 2 (b)).

Some works with empirical success inspire the proposal of DE, though we are the first one to derive theoretical characterization and leverage our theory to better those models as a return. SEAL [9] predicts links by reading out the representations of ego-networks of node-pairs. Although SEAL leverages a specific DE-2, the representations of those ego-networks are extracted via complex SortPooling [23]. However, we argue against such complex operations as DE-2 yields all the benefit of representations of node-pairs, as demonstrated by our experiments. PGNN [10] uses SPDs between each node and some anchor nodes to encode distance between nodes. As those encodings are not permutation invariant, PGNN holds worse inductive/generalization capability than our models.

Quite a few works targeted at revising neighbor aggregation procedure of WLGNN and thus are related to the DEA-GNN. However, none of them demystified their connection to DE or provided theoretical characterization. MixHop [47], PAGTN [31], MAT [17] essentially used $\zeta_{spd}(u|v)$ to change the way of aggregation (Eq. (4)) while GDC [32] and PowerGNN [48] used variants of $\zeta_{gpr}(u|v)$. MixHop, GDC and PowerGNN are evaluated for node classification while PAGTN and MAT are evaluated for graph classification. GDC claims that the aggregation based on $\zeta_{gpr}(u|v)$ does not help link prediction. However, we are able to show its empirical success for link prediction, as the key point missed by GDC is using DEs as extra node attributes (Appendix G.3). Note that as the above models are covered by DEA-GNN-1, their representation powers are all bounded by Theorem 3.7 according to Corollary 3.8.

## 5 Experiments

Extensive experiments[3] are conducted to evaluate our DE-GNN and DEA-GNN over three levels of tasks involving target node sets with sizes 1, 2 and 3 respectively: roles of nodes classification (Task 1), link prediction (Task 2), and triangle prediction (Task 3). Triangle prediction is to predict for any given subset of 3 nodes, $\{u, v, w\}$, whether links $uv$, $uw$, and $vw$ all exist. This task belongs to the more general class of higher-order network motif prediction tasks [49, 50] and has recently attracted much significance to [51–55]. We briefly introduce the experimental settings and save the details of the datasets and the model parameters to Appendix G.

**Dataset & Training.** We use the following six real graphs for the three tasks introduced above: Brazil-Airports (Task 1), Europe-Airports (1), USA-Airports (1), NS (2 & 3), PB (2), C.ele (2 & 3). For Task 1, the goal is to predict the passenger flow level of a given airport based solely on the flight traffic network. These airports datasets are chosen because the labels indicate the structural roles of nodes (4 levels in total from hubs to switches) rather than community identifiers of nodes as traditionally used [20, 21, 56]. For Tasks 2 & 3, the datasets were used by the strongest baseline [9], which consist of existing links/triangles plus the the same number of randomly sampled negative instances from those graphs. The positive test links/triangles are removed from the graphs during the training phase. For all tasks, we use 80%, 10%, 10% dataset splitting for training, validation, testing respectively. All the models are trained until loss converges and the testing performance of the best model on validation set is reported. We also report the experiments without validation sets that follow the original settings of the baselines [5, 9] in Appendix G.4.

| Data / Method | Nodes (Task 1): Average Accuracy | | | Node-pairs (Task 2): AUC | | | Node-triads (Task 3): AUC | |
|---|---|---|---|---|---|---|---|---|
| | Bra.-Airports | Eur.-Airports | USA-Airports | C.elegans | NS | PB | C.elegans | NS |
| GCN [20] | 64.55±4.18 | 54.83±2.69 | 56.58±1.11 | 74.03±0.99 | 74.21±1.72 | 89.78±0.99 | 80.94±0.51 | 81.72±1.50 |
| SAGE [21] | 70.65±5.33 | 56.29±3.21 | 50.85±2.83 | 73.91±0.32 | 79.96±1.40 | 90.23±0.74 | 84.72±0.40 | 84.06±1.14 |
| GIN [16] | 71.89±3.60† | 57.05±4.08 | 58.87±2.12 | 75.58±0.59 | 87.75±0.56 | 91.11±0.52 | 86.42±1.12† | 94.59±0.66† |
| Struc2vec [5] | 70.88±4.26 | 57.94±4.01† | 61.92±2.61† | 72.11±0.31 | 82.76±0.59 | 90.47±0.60 | 77.72±0.58 | 81.93±0.61 |
| PGNN [10] | N/A | N/A | N/A | 78.20±0.33 | 94.88±0.77 | 89.72±0.32 | 86.36±0.74 | 79.36±1.49 |
| SEAL [9] | N/A | N/A | N/A | 88.26±0.56† | 98.55±0.32† | 94.18±0.57† | N/A | N/A |
| DE-GNN-SPD | **73.28±2.47** | 56.98±2.79 | **63.10±0.68**\* | **89.37±0.17**\* | **99.09±0.79** | **94.95±0.37**\* | **92.17±0.72**\* | **99.65±0.40**\* |
| DE-GNN-LP | **75.10±3.80**\* | 58.41±3.20\* | **64.16±1.70**\* | 86.27±0.33 | 98.01±0.55 | 91.45±0.41 | 86.24±0.18 | **99.31±0.12**\* |
| DEA-GNN-SPD | **75.37±3.25**\* | 57.99±2.39\* | **63.28±1.59** | **90.05±0.26**\* | **99.43±0.63**\* | **94.49±0.24**\* | **93.35±0.65**\* | **99.84±0.14**\* |

Table 1: Performance in Average Accuracy and Area Under the ROC Curve (AUC) (mean in percentage ± 95% confidence level). † highlights the best baselines. \*, **bold font**, **bold font**\* respectively highlights the case where our models' performance exceeds the best baseline on average, by 70% confidence, by 95% confidence.

**Baselines.** We choose six baselines. GCN [20], GraphSAGE(SAGE) [21], GIN [16] are representative methods of WLGNN. These models use node degrees as initial features when attributes are not available to keep inductive ability. Struc2vec [5] is a kernel-based method, particularly designed for structural representations of single nodes. PGNN [10] and SEAL [9] are also GNN-based methods: PGNN learns node positional embeddings and is not inductive for node classification; SEAL is particularly designed for link prediction by using entire-graph representations of ego-networks of node-pairs. SEAL outperforms other link prediction approaches such as VGAE [57]. The node initial features for these two models are set as the inductive setting suggested in their papers. We tune parameters of all baselines (Appendix G.5) and list their optimal performance here.

**Instantiation of DE-GNN and DEA-GNN.** We choose GCN as the basic WLGNN and implement three variants of DE-GNN over it. Note that GIN could be a more powerful basis while we tend to keep our models simple. The first two variants of Eq. (2) give us DE-GNN-SPD and DE-GNN-LP. The former uses SPD-based one-hot vectors $\zeta_{sdp}$ as extra nodes attributes, and the latter uses the sequence of landing probabilities Eq. (2). Next, we consider an instantiation of Eq. (4), DEA-GNN-SPD that uses SPDs, $\zeta(u|v) = \zeta_{sdp}(u|v) \leq K$, to control the aggregation, which enables $K$-hop aggregation ($K = 2, 3$ and the better performance will be used). DEA-GNN-SPD uses SPD-based one-hot vectors as extra nodes attributes. Appendix G.3 provides thorough discussion on implementation of the three variants and another implementation that uses Personalized PageRank scores to control the aggregation. Experiments are repeated 20 times using different seeds and we report the average.

**Results** are shown in Table 1. Regarding the node-level task, GIN outperforms other WLGNNs, which matches the theory in [16, 26, 29]. Struc2vec is also powerful though it is kernel-based. DE-GNN's significantly outperform the baselines (except Eur.-Airport) which imply the power of DE-1's. Among them, landing probabilities (LP) work slightly better than SPDs as DE-1's.

Regarding node-pairs-level tasks, SEAL is the strongest baseline, as it is particularly designed for link prediction by using a special DE-2 plus a graph-level readout [23]. However, our DE-GNN-SPD performs even significantly better than SEAL: The numbers are close, but the difference is still significant; The decreases of error rates are always greater than 10% and achieve almost 30% over NS. This indicates that DE-2 is the key signal that makes SEAL work while the complex graph-level readout adopted by SEAL is not necessary. Moreover, our set-pooling form of DE-2 (Eq. (1)) decreases the dimension of DE-2 adopted in SEAL, which also betters the generalization of our models (See the detailed discussion in Appendix G.3). Moreover, for link prediction, SPD seems to be much better to be chosen as DE-2 than LP.

Regarding node-triads-level tasks, no baselines were particularly designed for this setting. We have not expected that GIN outperforms PGNN as PGNN captures node positional information that seems useful to predict triangles. We guess that the distortion of absolute positional embeddings learnt by PGNN may be the reason that limits its ability to distinguish structures with nodes in close positions: For example, the three nodes in a path of length two are close in position and the three nodes in a triangle are also close in position. However, this is not a problem for DE-3. We also conjecture that the gain based on DEs grows w.r.t. their orders (*i.e.*, $|S|$ in Eq. (1)). Again, for triangle prediction, SPD seems to be much better to be chosen as DE-3 than LP.

Note that DEA-GNN-SPD further improves DE-GNN-SPD (by almost 1% across most of the tasks). This demonstrates the power of multi-hop aggregation (Eq. (4)). However, note that DEA-GNN-SPD needs to aggregate multi-hop neighbors simultaneously and thus pays an additional cost of scalability.

## Broader Impact

This work proposes a novel angle to systematically improve the structural representation power of GNNs. We break from the convention that previous works characterize and further improve the power of GNNs by intimating different-order WL tests [26, 27, 30, 58]. As far as we know, we are the first one to provide non-asymptotic analysis of the power of the proposed GNN models. Therefore, the proof techniques of Theorems 3.3,3.7 may be expected to inspire new theoretical studies of GNNs and further better the practical usage of GNNs. Moreover, our models have good scalability by avoiding using the framework of WL tests, as higher-order WL tests are not able to leverage the sparsity of graphs. To be evaluated over extremely large graphs [59], our models can be simply trimmed and work on the ego-networks sampled with a limited size around the target node sets, just as the strategy adopted by GraphSAGE [21] and GraphSAINT [60]. **Therefore, our work may motivate practitioners to design and deploy more powerful GNNs in industrial pipelines to benefit the society.**

Distance encoding unifies the techniques of many GNN models [9, 17, 31, 32, 47] and provides a extremely general framework with clear theoretical characterization. In this paper, we only evaluate four specific instantiations over three levels of tasks. However, there are some other interesting instantiations and applications. For example, we expect a better usage of PageRank scores as edge attributes (Eq. (4)). Currently, our instantiation DEAGNN-PR simply uses those scores as weights in a weighted sum to aggregate node representations. We also have not considered any attention-based mechanism over DEs in aggregation while it seems to be useful [17, 22]. Researchers may try these directions in a more principled manner based on this work. Our approaches may also help other tasks based on structural representation learning, such as graph-level classification/regression [14, 16, 23, 26, 27] and subgraph counting [58], which correspond to **many applications with wide societal impact including drug discovery and structured data analysis.**

There are also two important implications coming from the observations of this work. First, Theorem 3.7 and Corollary 3.8 show the limitation of DE-1 over distance regular graphs, including the cases when DE-1's are used as node attributes or controllers of message aggregation. As distance regular graphs with the same intersection array have the important co-spectral property [35], we guess that DE-1 is a bridge to connect GNN frameworks to spectral approaches, two fundamental approaches in graph-structured data processing. This point sheds some light on the question left in [30] while more rigorous characterization is still needed. Second, as observed in the experiments, higher-order DE's induce larger gains as opposed to WLGNN, while Theorem 3.3 is not able to characterize this observation as the probability $1 - o(\frac{1}{n})$ does not depend on the size $p$. We are sure that the probabilistic quantization in Theorem 3.3 is not tight, so it is interesting to see how such probability depends on $p$ by deriving tighter bounds.

**We are not aware of any societal disadvantages of this research. Our experiments also do not leverage biases in the data.** We choose those datasets based on two rules that are irrelevant to ethics: (1) The labels for evaluation are graph-structured related; (2) Those datasets were used in some baselines so that we provide fair comparison. Regarding the point (1), this is the reason that we do not use the datasets, such as Cora, Citeseer and Pubmed [56], whose node labels that indicate the community belongings of nodes instead of the structural function of nodes. However, it is interesting to evaluate distance encoding techniques over those datasets with community-related labels in the future.

## Acknowledgements

The authors would like to thank Weihua Hu for raising the preliminary concept of distance encoding that initializes the investigation. The authors also would like to thank Jiaxuan You and Rex Ying for their insightful comments during the discussion. The authors also would thank Baharan Mirzasoleiman and Tailin Wu for their suggestions on the paper writing. The authors also would like to thank the NeurIPS reviewers for their insightful observations and actionable suggestions to improve the manuscript. This research has been supported in part by Purdue CS start-up, NSF CINES, NSF HDR, NSF Expeditions, NSF RAPID, DARPA MCS, DARPA ASED, ARO MURI, Stanford Data Science Initiative, Wu Tsai Neurosciences Institute, Chan Zuckerberg Biohub, Amazon, Boeing, Chase, Docomo, Hitachi, Huawei, JD.com, NVIDIA, Dell. J. L. is a Chan Zuckerberg Biohub investigator.

## Footnotes

[1] We follow the terminology 2-WL test in [39], which refines representations of node-pairs iteratively and is proved to be more powerful than 1-WL test. This 2-WL test is termed 2-WL' test in [40] or 2-FWL test in [27]. A brief introduction of higher-order WL tests can be found in Appendix H.

[2] Actually, the 2-WL test may not distinguish edges in a special class of DRGs, termed strongly regular graphs (SRG) [39]. A connected SRG is a DRG with diameter constrained as 2 [41].

[3]The code to evaluate our model can be downloaded from https://github.com/snap-stanford/distance-encoding.

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
