[Supplementary Material]

# Distance Encoding: Design Provably More Powerful Neural Networks for Graph Representation Learning

**Pan Li**
Department of Computer Science
Purdue University
panli@purdue.edu

**Yanbang Wang**
Department of Computer Science
Stanford University
ywangdr@cs.stanford.edu

**Hongwei Wang**
Department of Computer Science
Stanford University
hongweiw@cs.stanford.edu

**Jure Leskovec**
Department of Computer Science
Stanford University
jure@cs.stanford.edu

## Abstract

Learning representations of sets of nodes in a graph is crucial for applications ranging from node-role discovery to link prediction and molecule classification. Graph Neural Networks (GNNs) have achieved great success in graph representation learning. However, expressive power of GNNs is limited by the 1-Weisfeiler-Lehman (WL) test and thus GNNs generate identical representations for graph substructures that may in fact be very different. More powerful GNNs, proposed recently by mimicking higher-order-WL tests, only focus on representing entire graphs and they are computationally inefficient as they cannot utilize sparsity of the underlying graph. Here we propose and mathematically analyze a general class of structure-related features, termed Distance Encoding (DE). DE assists GNNs in representing any set of nodes, while providing strictly more expressive power than the 1-WL test. DE captures the distance between the node set whose representation is to be learned and each node in the graph. To capture the distance DE can apply various graph-distance measures such as shortest path distance or generalized PageRank scores. We propose two ways for GNNs to use DEs (1) as extra node features, and (2) as controllers of message aggregation in GNNs. Both approaches can utilize the sparse structure of the underlying graph, which leads to computational efficiency and scalability. We also prove that DE can distinguish node sets embedded in almost all regular graphs where traditional GNNs always fail. We evaluate DE on three tasks over six real networks: structural role prediction, link prediction, and triangle prediction. Results show that our models outperform GNNs without DE by up-to 15% in accuracy and AUROC. Furthermore, our models also significantly outperform other state-of-the-art methods especially designed for the above tasks.

## 1 Introduction

Graph representation learning aims to learn representation vectors of graph-structured data [1]. Representations of node sets in a graph can be leveraged for a wide range of applications, such as discovery of functions/roles of nodes based on individual node representations [2–6], link or link type prediction based on node-pair representations [7–10] and graph comparison or molecule classification based on entire-graph representations [11–17].

Graph neural networks (GNNs), inheriting the power of neural networks [18], have become the *de facto* standard for representation learning in graphs [19]. Generaly, GNNs use message passing procedure over the input graph, which can be summarized in three steps: (1) Initialize node representations with their initial attributes (if given) or structural features such as node degrees;

| **WLGNN-$p$ to represent $\mathcal{T} = (S, \mathbf{A})$, $\|S\| = p$** |
|---|
| Initialize: For all $v \in V$, $h_v^{(0)} = \mathbf{A}_{vv}$ |
| For layers $l = 0, 1, ..., L - 1$ and all $v \in V$, do: |

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

# Appendix

## A  Proof of Universal Approximate Theorem for Structural Representation

We restate Theorem 2.6: If the structural representation $\Gamma$ can distinguish any two non-isomorphic tuples $\mathcal{T}^{(1)}$ and $\mathcal{T}^{(2)}$ in $\Omega_p$, then for any invariant function $f : \Omega_p \to \mathbb{R}$, $f$ can be universally approximated by $\Gamma$ via a 3-layer feed-forward neural network with ReLU as rectifiers, as long as

- The feature space $\mathcal{A}$ is compact.
- $f(S, \cdot)$ is continuous over $\mathcal{A}$ for any $S \in \mathcal{P}_p(V)$.

*Proof.* This result is a direct generalization of Theorem 4 [30]. Specifically, we extend the statement of representing graphs featured by $\mathbf{A}$ to that of representing structures featured by $(S, \mathbf{A})$.

Recall the original space $\mathcal{A} \subset \mathbb{R}^{n \times n \times k}$. We define a space $\mathcal{A}' \subset \mathbb{R}^{n \times n \times (k+1)}$: For any $\mathbf{A}' \in \mathcal{A}'$, its slice in the first $k$ dimensions of 3-rd mode, *i.e.*, $\mathbf{A}'_{\cdot,\cdot,1:k}$, is in $\mathcal{A}$ and the slice corresponds to the last dimension of 3-rd mode, *i.e.*, $\mathbf{A}'_{\cdot,\cdot,k+1}$ is a diagonal matrix where the diagonal components could be only 0 or 1. Then, we may build a bijective mapping between $\mathbf{A}' \in \mathcal{A}'$ and $(S, \mathbf{A}) \in \Omega_p$ by

$$\mathbf{A}'_{\cdot,\cdot,1:k} = \mathbf{A}, \quad \mathbf{A}'_{u,u,k+1} = 1 \text{ if } u \in S \text{ or } 0 \text{ if } u \notin S$$

As $\mathcal{A}$ is compact in $\mathbb{R}^{n \times n \times k}$ and we have only finite possible choices of $\mathbf{A}'_{\cdot,\cdot,k+1}$, actually $\binom{n}{|S|}$, the space $\mathcal{A}'$ is compact in $\mathbb{R}^{n \times n \times (k+1)}$.

Then, we may transfer all definitions from $\Omega_p$ to $\mathcal{A}'$. Specifically, the structural representation $\Gamma$ that distinguishes any two non-isomorphic tuples $\mathcal{T}^{(1)}$ and $\mathcal{T}^{(2)}$ in $\Omega_p$ defines $\Gamma' : \mathcal{A}' \to \mathbb{R}^d$ that distinguishes any two non-isomorphic tensors $\mathbf{A}'^{(1)}$ and $\mathbf{A}'^{(2)}$ in $\mathcal{A}'$, as $\mathcal{T}^{(i)}$ and $\mathbf{A}'^{(i)}$ form a bijective mapping for $i = 1, 2$. Moreover, one invariant function $f : \Omega_p \to \mathbb{R}$ also defines another invariant function $f' : \mathcal{A}' \to \mathbb{R}$, as $\mathcal{T}^{(i)}$ and $\mathbf{A}'^{(i)}$ form a bijective mapping for $i = 1, 2$.

Suppose the original metric over $\mathcal{A}$ is denoted by $\mathcal{M} : \mathcal{A} \times \mathcal{A} \to \mathbb{R}_{\geq 0}$. Define a metric over $\mathcal{A}'$ as

$$\mathcal{M}'(\mathbf{A}'^{(1)}, \mathbf{A}'^{(2)}) = \mathcal{M}(\mathbf{A}'^{(1)}_{\cdot,\cdot,1:k}, \mathbf{A}'^{(2)}_{\cdot,\cdot,1:k}) + \sum_{u \in V} \mathbb{1}_{\mathbf{A}'^{(1)}_{u,u,k+1} \neq \mathbf{A}'^{(2)}_{u,u,k+1}}.$$

Then, it is easy to show that we have the following lemma based on the definition of continuity.

**Lemma A.1.** If $f(S, \cdot)$ is continuous over $\mathcal{A}$ for any $S \in \mathcal{P}_p(V)$ with respect to $\mathcal{M}$, then $f'$ is continuous over $\mathcal{A}'$ with respect to $\mathcal{M}'$.

Now, we only need to use Theorem 4 [30] to prove the statement. Actually the dimensions of $\Gamma'$ overall forms a collection of one-dimensional functions $\Xi = (\Gamma'[i])_{i \in [d]}$, where $\Gamma'[i]$ is the $i$th component of $\Gamma' \in \mathbb{R}^d$. According to the definition of $\Gamma'$, we know $\Xi$ distinguishes all the non-isomorphic $\mathbf{A}'^{(1)}$ and $\mathbf{A}'^{(2)}$ in $\mathcal{A}'$. Moreover, because $\mathcal{A}'$ is compact and $f'$ is continuous over $\mathcal{A}'$, Theorem 4 [30] shows that the arbitrary invariant function $f'$ defined on $\mathcal{A}'$ can be universally approximated by $\Xi$ via 3-layer feed-forward neural networks with ReLu as rectifiers. Recall $f$ and $f'$ are bijective, and $\Gamma$, $\Gamma'$, and $\Xi$ are mutually bijective. Therefore, we claim that $f$ can be universally approximated $\Gamma$ via 3-layer feed-forward neural networks with ReLu as rectifiers. $\square$

## B  Proof for The Power of WLGNN for Structural Representation

We restate Theorem 2.7: Consider two tuples $\mathcal{T}_1 = (S^{(1)}, \mathbf{A}^{(1)})$ and $\mathcal{T}_2 = (S^{(2)}, \mathbf{A}^{(2)})$ in $\Omega_p$. If $\mathcal{T}_1, \mathcal{T}_2$ cannot be distinguished by the 1-WL test, then the corresponding outputs of WLGNN satisfy $\Gamma(\mathcal{T}_1) = \Gamma(\mathcal{T}_2)$. On the other side, if they can be distinguished by the 1-WL test and suppose aggregation operations (AGG) and feed-forward neural networks $f_1$, $f_2$ are all injective mappings, then a large enough number of layers $L$, the outputs of WLGNN satisfy $\Gamma(\mathcal{T}_1) \neq \Gamma(\mathcal{T}_2)$.

*Proof.* There is no fundamental difficulty to generalize the results from the case of graph representation to that of structural representation, because the only difference according to WLGNN is the final

readout step (AGG($\cdot$)), which works on a subset of nodes instead of the entire node set. Therefore, the same logic of proofs of Lemma 2 and Theorem 3 in [16] can be directly applied for structural representation learning with little revision. $\square$

## C Proof for The Power of DE — Theorem 3.3

We restate Theorem 3.3: Given two fixed-sized sets $S^{(1)}, S^{(2)} \subset V, |S^{(1)}| = |S^{(2)}| = p$. Consider two tuples $\mathcal{T}^{(1)} = (S^{(1)}, \mathbf{A}^{(1)})$ and $\mathcal{T}^{(2)} = (S^{(2)}, \mathbf{A}^{(2)})$ in the most difficult setting when features $\mathbf{A}^{(1)}$ and $\mathbf{A}^{(2)}$ are only different in graph structures specified by $A^{(1)}$ and $A^{(2)}$ respectively. Suppose $A^{(1)}$ and $A^{(2)}$ are uniformly independently sampled from all r-regular graphs over $V$ where $3 \leq r < (2 \log n)^{1/2}$. Then, for any constant $\epsilon > 0$, there exist a proper DE-GNN-$p$ with layers $L < (\frac{1}{2} + \epsilon)\frac{\log n}{\log(r-1)}$, using DE-$p$ $\zeta(u|S^{(1)}), \zeta(u|S^{(2)})$ for all $u \in V$ such that with probability $1 - o(n^{-1})$, its outputs $\Gamma(\mathcal{T}^{(1)}) \neq \Gamma(\mathcal{T}^{(2)})$. Specifically, $f_3$ can be simply chosen as SPD, $i.e.$, $\zeta(u|v) = \zeta_{spd}(u|v)$. The big-O notation is with respect to $n$.

*Proof.* To prove the statement, we only need to prove the case that $|S^{(1)}| = |S^{(2)}| = 1$, $\zeta(u|v) = \zeta_{spd}(u|v)$ because of the following lemma.

**Lemma C.1.** Suppose the statement is true when $|S^{(1)}| = |S^{(2)}| = 1$, $\zeta(u|v) = \zeta_{spd}(u|v)$. Then, the statement is also true for the case when $|S^{(1)}| = |S^{(2)}| = p > 1$ for some fixed $p$, and $\zeta(u|v)$ is a neural network fed with the list of landing probabilities.

*Proof.* We first focus on the case when DE is chosen as SPD, $i.e.$, $\zeta(u|v) = \zeta_{spd}(u|v)$. We want to use the results of the case $|S^{(1)}| = |S^{(2)}| = 1$ to prove that of the case $|S^{(1)}| = |S^{(2)}| > 1$. Suppose $|S^{(1)}| = |S^{(2)}| > 1$. We choose an arbitrary node from $S^{(1)}$, say $w_1$. As we assume the statement is true for the single node case, for any node in $S^{(2)}$, say $w_2$, DE-GNN with DEs $\zeta_{spd}(u|w_1)$, $\zeta_{spd}(u|w_2)$ is able to distinguish two tuples $(w_1, \mathbf{A}^{(1)})$ and $(w_2, \mathbf{A}^{(2)})$, with probability at least $1 - o(n^{-1})$. Given that the space of SPD is countable and AGG in Eq. (1) is injective, $\zeta_{spd}(u|S^{(1)})$ and $\zeta_{spd}(u|S^{(2)})$ are different if $\zeta_{spd}(u|w_1)$ is different from any $\zeta_{spd}(u|w_2)$ $(w_2 \in S^{(2)})$, which happens with probability at least $1 - |S^{(2)}|o(n^{-1}) = 1 - o(n^{-1})$. Therefore, DE-GNN with DEs $\zeta_{spd}(u|S^{(1)})$, $\zeta_{spd}(u|S^{(2)})$ is also able to distinguish two tuples $(w_1, \mathbf{A}^{(1)})$ and $(w_2, \mathbf{A}^{(2)})$, with probability at least $1 - o(n^{-1})$. Based on the union bound, we know that DE-GNN with DEs $\zeta_{spd}(u|S^{(1)})$, $\zeta_{spd}(u|S^{(2)})$ is able to distinguish two tuples $(w_1, \mathbf{A}^{(1)})$ and $(v, \mathbf{A}^{(2)})$ for any $v \in S^{(2)}$, with probability at least $1 - |S^{(2)}|o(n^{-1}) = 1 - o(n^{-1})$. Therefore, we prove the capability to generalize the result from the single node case to the multiple node cases.

Now, let us generalize the result from $\zeta_{spd}(u|v)$ to arbitrary $\zeta(u|v)$ represented by neural networks fed with the list of landing probabilities. As $\zeta_{spd}(u|v)$ is indeed a function of the list of landing probabilities (Eq. (2)), the general $\zeta(u|v)$ should have stronger discriminatory power unless neural networks cannot provide a good mapping from the list of landing probabilities to SPD. However, we do not have to worry this because the list of landing probabilities fortunately lies a countable space for unweighted graphs: 1) The dimension of this list is countable (finite in practice); 2) Each component of this list is always a rational number if the graph is unweighted. According to our assumption, $f_3$ is allowed to have an injective mapping over the list of landing probabilities. Therefore, neural networks on the list of landing probabilities will not decrease the representation power that is just based on $\zeta_{spd}(u|v)$. $\square$

Keep in mind that Lemma C.1 could be loose. We leave the tight probability bound for the $p > 2$ case in the future.

**Outline:** From now on, we focus on the single node case, $i.e.$ $|S^{(1)}| = |S^{(2)}| = 1$, with SPD as the DE-1 ($f_3$), $i.e.$, $\zeta(u|v) = \zeta_{spd}(u|v)$. Without loss of generality, we suppose $S^{(1)} = S^{(2)} = \{u\}$. As SPD is countable, there exists a proper DE-GNN that guarantees that all the operations are injective, which follows the basic condition used in [16]. Because of the iterative procedure of DE-GNN and all mappings are injection, the label of node $u$ only depends on the subtree with depth $L$ rooted at $u$ (See the illustration of the subtree rooted at a given node in Fig. 3). Recall $L$ is the number of layers

Figure 3: The subtree rooted at a node: In the left two graphs, the black nodes are the target nodes who structural representations are to be learnt. Different colors of the nodes correspond to different SPDs with respect to the target nodes. The right two trees correspond to the subtrees rooted at these two target nodes respectively. DE-GNN essentially works from bottom to top along these subtrees to obtain the representations of the target nodes. Different types of subtrees yield different representations based on a proper DE-GNN. In this example, these two target nodes are all from 3-regular graphs so they cannot be distinguished via WLGNN without DE-1's or the informative node/edge attributes. However, with DE-1's, we can see the corresponding subtrees of these two nodes are different and the difference appears in the second layer.

in DE-GNN. Therefore, we only need to show that $A^{(1)}$ and $A^{(2)}$, which are uniformly sampled $r$-regular graphs, with probability at most $o(n^{-1})$, have the same subtrees rooted at $u$ given SPDs as initial node labels. To show this, our proof contains four steps. Note that all through the following proof, we assume that $n$ is very large and $\epsilon$ is any small positive constant that is independent from $n$.

- We first explain that we are able to work on the configuration model of $r$-regular graphs proposed in [61] which associates uniform measure over all $r$-regular graphs. Given the condition $r < (2 \log n)^{1/2}$, there are a large portion ($\Omega(n^{-1/2})$) of all the graphs generated by this model are simple (without self-loops and multi-edges) $r$-regular graphs [61]. Since the configuration model alleviates the difficulty to analyze the dependence between edges in $r$-regular graphs, we consider the graphs generated by the configuration model for the next two steps.

- Suppose the set of nodes that are associated with SPD= $k$ from $u$ is denoted by $Q_k$, and the number of edges that connect the nodes in $Q_k$ and those in $Q_{k+1}$ is denoted by $p_k$. We prove that with probability $1 - o(n^{-\frac{3}{2}})$, for all $k \in (\frac{\epsilon}{5} \frac{\log n}{\log(r-1)} + 1, (\frac{2}{3} - \epsilon) \frac{\log n}{\log(r-1)})$, $|Q_k| \geq (r - 1 - \epsilon)^{k-1}$ and $p_k \geq (r - 1 - \epsilon)|Q_k|$ based on the configuration model.

- Next, we define the edge configuration between $Q_k$ and $Q_{k+1}$ as a list $C_k = (a_{1,k}, a_{2,k}, ...)$ where $a_{i,k}$ denotes the number of nodes in $Q_{k+1}$ of which each has exactly $i$ edges from $Q_k$. We prove that for each $k \in (\frac{1}{2} \frac{\log n}{\log(r-1-\epsilon)}, \frac{4}{7} \frac{\log n}{\log(r-1-\epsilon)})$, as the edges between $Q_k$ and $Q_{k+1}$ are so many, there are too much randomness that makes each type of edge configuration $C_k$ appear with only limited probability $\mathbb{P}(C_k) = O(\frac{n^{1/2}}{p_k})$. Recall that $p_k$ is defined in step 2. Then, given any $\epsilon \frac{\log n}{\log(r-1-\epsilon)}$ many $k$'s, the probability that $A^{(1)}$ and $A^{(2)}$ have all

the same edge configurations for these $k$'s is bounded by $\Pi_k \mathbb{P}(C_k) \sim o(n^{-\frac{3}{2}})$. Therefore, we only need to consider edge configurations for $k \in (\frac{1}{2} \frac{\log n}{\log(r-1-\epsilon)}, (\frac{1}{2} + \epsilon) \frac{\log n}{\log(r-1-\epsilon)})$ to distinguish $A^{(1)}$ and $A^{(2)}$, as this will give us $1 - o(n^{-3}2)$ probability to succeed.

- Since there are at least $\Omega(n^{-1/2})$ of all the graphs generated by the configuration model that are simple $r$-regular graphs, and there are at most $o(n^{-\frac{3}{2}})$ probability that $A^{(1)}$ and $A^{(2)}$ share the same subtrees rooted at $u$, there are at most $o(n^{-\frac{3}{2}}/n^{-1/2}) = o(n^{-1})$ probability that $A^{(1)}$ and $A^{(2)}$ are simple $r$-regular graphs and share the same subtrees rooted at $u$, which concludes the proof.

**Step 1:** We first introduce the configuration model proposed in [61] for $r$-regular graphs of $n$ nodes. Suppose we have $n$ sets of items, $W_u$, $u \in [n]$, where each set corresponds to one node in $[n]$. Each set $W_u$ has $r$ items. Now, we randomly partition all these $nr$ items into $\frac{nr}{2}$ pairs. Then, each partitioning result corresponds to a r-regular graph: if a pair contains items from $W_u$ and $W_v$, then there is an edge between nodes $u$ and $v$ in the graph. Note that such partitioning results may render self-loops and multi-edges. Of course, we would like to consider only simple graphs which do not have self-loops and multi-edges. For this, the theory in [61] shows that for all these $r$-regular graphs, if $r < (2 \log n)^{1/2}$, there are about $\exp(-\frac{r^2-1}{4})$ portion among them, *i.e.*, $\Omega(n^{-1/2})$, which are simple graphs.

**Step 2:** Now, we consider a graph that is uniformly sampled from the configuration model. Recall that the set of nodes that are associated with SPD= $k$ from $u$ is denoted by $Q_k$, and the number of edges that connect the nodes in $Q_k$ and those in $Q_{k+1}$ is denoted by $p_k$. Now, we prove that there exists a small constant $\epsilon > 0$, such that with probability $1 - o(n^{-\frac{3}{2}})$, for all $k \in (\frac{\epsilon}{5} \frac{\log n}{\log(r-1)} + 1, (\frac{2}{3} - \epsilon) \frac{\log n}{\log(r-1)})$, $|Q_k| \geq (r-1-\epsilon)^{k-1}$ and $p_k \geq (r-1-\epsilon)|Q_k|$. We prove an even stronger lemma that gives the previous argument via a union bound and doing induction over all $k \in (\frac{\epsilon}{5} \frac{\log n}{\log(r-1)} + 1, (\frac{2}{3} - \epsilon) \frac{\log n}{\log(r-1)})$.

**Lemma C.2.** There exists a small constant $\epsilon > 0$, with probability $1 - O(n^{-2+\epsilon})$, such that: 1) For any $k < (\frac{2}{3} - \epsilon) \frac{\log n}{\log(r-1)}$, if $|Q_k| \geq n^{\epsilon/5}$, $|Q_{k+1}| \geq p_k - |Q_k|^{1/2}$ and $p_k \geq (r-1)|Q_k| - |Q_k|^{1/2}$; 2) When $k = \lceil \frac{\epsilon}{5} \frac{\log n}{\log(r-1)} \rceil + 1$, $|Q_k| \geq (r-1)^{k-1} = n^{\epsilon/5}$.

*Proof.* We consider the following procedure to generate the graph based on the configuration model. Recall the node $u$ is the target node, or the node with $SPD = 0$ to the target node. We start from generating the edges attached to this node. We start from the set $W_u$ and generate the $r$ pairs with at least one item in $W_u$. Then, we have all the nodes in $Q_1$. Based on the set $\cup_{v \in Q_1} W_v$, we generate all the $(r-1)|Q_1|$ pairs with at least one item in $\cup_{v \in Q_1} W_v$, and we have all the nodes in $Q_2$. The procedure goes on so on and so forth, from $Q_k$ to $Q_{k+1}$.

Now, we prove 1). First, we prepare some inequalities. We have $|Q_k| \leq r(r-1)^{k-1} < n^{2/3-\epsilon}$ by the assumption on $k$. For $i \geq \lceil |Q_k|^{1/2} \rceil$, we have

$$\frac{|Q_k| \cdot (r-1)|Q_k|}{n \cdot i} \leq n^{-\epsilon} \tag{5}$$

Moreover, Recall $|Q_k| < n^{2/3-\epsilon}$. As $|Q_k| \geq n^{\epsilon/5}$, then

$$\left( \frac{e(r-1)|Q_k|^{3/2}}{n} \right)^{\lceil |Q_k|^{1/2} \rceil} = O(n^{-2+\epsilon}). \tag{6}$$

This inequality is very crude.

Now, we go back to prove the bound for $p_k$. Recall the definition of $p_k$ that is the number of edges between $Q_k$ and $Q_{k+1}$. Then, the number of edges that are generated with both end-nodes are in $Q_k$ is $(r-1)|Q_k| - p_k$. As we suppose the edges are generated sequentially, the probability to generate an edge whose two end-nodes are in $Q_k$ is upper bounded by $\frac{(r-1)|Q_k|}{r(n - \sum_{j=0}^{k} |Q_k|)} < \frac{|Q_k|}{n}$ where we use

$\sum_{j=0}^{k} |Q_j| \leq r(r-1)^k = O(n^{2/3})$. Then, the probability that $(r-1)|Q_k| - p_k > |Q_k|^{1/2}$ is upper bounded by (just summing over all possible $(r-1)|Q_k| - p_k = i > |Q_k|^{1/2}$)

$$\sum_{i=\lceil |Q_k|^{1/2} \rceil}^{(r-1)|Q_k|} \left[ \frac{|Q_k|}{n} \right]^i \binom{(r-1)|Q_k|}{i} \overset{(5)}{<} \left[ \frac{|Q_k|}{n} \right]^{\lceil |Q_k|^{1/2} \rceil} \binom{(r-1)|Q_k|}{\lceil |Q_k|^{1/2} \rceil} \sum_{i \geq 0} n^{-i\epsilon}$$

$$< c_1 \left[ \frac{|Q_k|}{n} \right]^{\lceil |Q_k|^{1/2} \rceil} \binom{(r-1)|Q_k|}{\lceil |Q_k|^{1/2} \rceil}$$

$$< c_2 \left[ \frac{|Q_k|}{n} \right]^{\lceil |Q_k|^{1/2} \rceil} \left[ \frac{e(r-1)|Q_k|}{\lceil |Q_k|^{1/2} \rceil} \right]^{\lceil |Q_k|^{1/2} \rceil} \overset{(6)}{=} O(n^{-2+\epsilon})$$

where $c_1, c_2$ are constants, the numbers above the equality/inequality signs refer to which equations are used.

Next, we prove the bound for $|Q_{k+1}| \geq p_k - |Q_k|^{1/2}$. Again, if the edges are generated sequentially, $p_k - Q_{k+1}$ indicates the number of edges whose end-nodes in $Q_{k+1}$ also belong to other edges that has been generated between $Q_k$ and $Q_{k+1}$. The probability of this edge is upper bounded by $\frac{|Q_{k+1}|}{r(n-\sum_{j=0}^{k+1} |Q_j|)} < \frac{(r-1)|Q_k|}{r(n-\sum_{j=0}^{k+1} |Q_j|)} \leq \frac{|Q_k|}{n}$. Then, the probability that $p_k - Q_{k+1} > |Q_k|^{1/2}$ is again upper bounded by (just summing over all possible $p_k - Q_{k+1} = i > |Q_k|^{1/2}$)

$$\sum_{i=\lceil |Q_k|^{1/2} \rceil}^{(r-1)|Q_k|} \left[ \frac{|Q_k|}{n} \right]^i \binom{(r-1)|Q_k|}{i} = O(n^{-2+\epsilon})$$

Till now, we have proved the statement 1).

Now, we prove the statement 2). Actually, at the time when $Q_k$ is generated, the number of edges having been generated is at most $r(r-1)^k - 2$. These edges cover at most $r(r-1)^k - 1$ nodes. When $k = \lceil \frac{\epsilon}{5} \frac{\log n}{\log(r-1)} \rceil + 1$, we claim that with probability $1 - O(n^{-2+\epsilon})$, at most 1 edge among these edges when generated is not connected to a new node. This is because if there are more than 1 such edges, the probability is at most (by summing over $i$ such edges)

$$\sum_{i=2}^{r(r-1)^k} \left[ \frac{r(r-1)^k}{n-r(r-1)^k} \right]^i \binom{r(r-1)^k}{i} \leq c_3 \left( \frac{r^2 n^{\epsilon/5}}{n} \right)^2 \binom{r^2 n^{\epsilon/5}}{2} = O(n^{-2+\epsilon}).$$

We use this result to give a lower bound of $|Q_k|$ for $k = \lceil \frac{\epsilon}{5} \frac{\log n}{\log(r-1)} \rceil + 1$. Because there is at most 1 edge when generated do not connect to a new node. The worst case appears when two items in $W_u$ are mutually connected, which leads to $|Q_1| \geq r - 2 \geq 1$. All edges after $Q_1$ is generated are connected to new nodes and furthermore $|Q_k| \geq (r-1)^{k-1} \geq n^{\epsilon/5}$. $\qquad \square$

**Step 3:** We start to consider the edge configuration between $Q_k$ and $Q_{k+1}$ for $k \in (\frac{1}{2} \frac{\log n}{\log(r-1-\epsilon)}, \frac{4}{7} \frac{\log n}{\log(r-1-\epsilon)})$. We focus our attention on the graphs that satisfy the properties developed in Step 2, which, as demonstrated in Step 2, are with high probability $1 - o(n^{-3/2})$. For those graphs, we know that for $k \in (\frac{1}{2} \frac{\log n}{\log(r-1-\epsilon)}, \frac{4}{7} \frac{\log n}{\log(r-1-\epsilon)})$, $p_k \geq (r-1-\epsilon)|Q_k| \geq (r-1-\epsilon)^k \geq n^{1/2}$ and $p_k \leq r(r-1)^k < n^{2/3-\epsilon}$. Moreover, $\sum_{j=1}^{k} |Q_j| \leq (r-1)|Q_k| = o(n)$ and therefore at the time when $Q_k$ is generated, there are still $q_k = n - o(n) = \Theta(n)$ nodes that have not been connected.

Recall that we define the edge configuration between $Q_k$ and $Q_{k+1}$ is a list $C_k = (a_{1,k}, a_{2,k}, ...)$ where $a_{i,k}$ means the number of nodes in $Q_{k+1}$ of which each has exactly $i$ edges from $Q_k$. According to the definition of $C_k$, it satisfies

$$\sum_{i=1}^{r} i \times a_{i,k} = p_k \tag{7}$$

Note that if DE-GNN cannot distinguish $(u, A^{(1)})$ and $(u, A^{(2)})$, then $A^{(1)}$ and $A^{(2)}$ must share the same edge configuration between $Q_k$ and $Q_{k+1}$. Otherwise, after one iteration, the intermediate

representation of nodes in $Q_{k+1}$ are different between $A^{(1)}$ and $A^{(2)}$. Such difference will be propagated to $u$ later. To bound the probability that $A^{(1)}$ and $A^{(2)}$ must share the same edge configuration between $Q_k$ and $Q_{k+1}$, for simplicity, we consider the probability of $C_k$ given the number of edges between $Q_k$ and $Q_{k+1}$, i.e., $p_k$ and the number remaining nodes, i.e., $q_k = [n]/\cup_{i=1}^{k} Q_i = \Theta(n)$. We are to derive a upper bound of $\mathbb{P}(C_k)$ based on the configuration model in the following lemma.

**Lemma C.3.** Suppose $p_k \in [n^{1/2}, n^{2/3-\epsilon}]$ and $q_k = \Theta(n)$. Consider the configuration model to generate edges: there are $p_k$ edges that correspond to two items that are one in $\cup_{v \in Q_k} W_v$ and one among the rest $q_k r$ items. Then, for any possible edge configuration $C_k$ obtained based on this generating procedure, $\mathbb{P}(C_k) \le c_5 \dfrac{q_k^{1/2}}{p_k}$ for some constant $c_5$.

*Proof.* First, for the configuration $C_k = (a_{1,k}, a_{2,k}, ...)$, we claim that the most probable $C_k$ is achieved when $a_{i,k} = 0$ for $i \ge 3$. We prove this statement via the adjustment method: We fix the value of $a_{1,k} + i a_{i,k}$ and all the other $a_{i',k}$'s. We compare the probability of $(a_{1,k} = x, a_{i,k} = y)$ and that of $(a_{1,k} = x + i, a_{i,k} = y - 1)$. Because $x, y \le p_k = O(n^{2/3-\epsilon})$, for some constant $c_6$, we have

$$\frac{\mathbb{P}(a_{1,k} = x, a_{i,k} = y)}{\mathbb{P}(a_{1,k} = x+i, a_{i,k} = y-1)} = \frac{\binom{q_k}{x} r^x \binom{q_k - x}{y} \binom{r}{i}^y}{\binom{q_k}{x+i} r^{x+i} \binom{q_k - x - i}{y-1} \binom{r}{i}^{y-1}} \le c_6 \frac{x^i}{q_k^{i-1}} \le c_6 \frac{x^3}{q_k^2} < 1.$$

Therefore, we only need to consider the case when $a_{1,k}, a_{2,k} > 0$ so $a_{1,k} + 2a_{2,k} = p_k$. Define a function $g(x)$ to denote the probability of the edge configuration $(a_{1,k} = p_k - 2y, a_{2,k} = y)$. We compare $g(y)$ and $g(y+1)$

$$\frac{g(y)}{g(y+1)} = \frac{\binom{q_k}{p_k - 2y} r^{p_k - 2y} \binom{q_k - p_k + 2y}{y} \binom{r}{2}^y}{\binom{q_k}{p_k - 2y - 2} r^{p_k - 2y - 2} \binom{q_k - p_k + 2y + 2}{y+1} \binom{r}{2}^{y+1}} = \frac{2r}{(r-1)} \frac{(y+1)(q_k - p_k + y + 1)}{(p_k - 2y)(p_k - 2y - 1)}. \quad (8)$$

Consider the choice $y = y^*$ to make $g(y^*)/g(y^*+1) \ge 1$ while $g(y^*-1)/g(y^*) \le 1$ that corresponds to $g(y^*) = \max_y g(y)$. Then, we must have $y^* = o(p_k)$ and otherwise $g(y^*-1)/g(y^*) > 1$ because $q_k = \Theta(n)$. As $y^* = o(p_k)$ and $p_k = o(q_k)$, $y^*$ according to Eq. 8 is about $\frac{(r-1)p_k^2}{2rq_k}$ by setting Eq. 8 $= 1$. We define $y_0 = \frac{(r-1)p_k^2}{2rq_k}$. Consider $y_0 + \delta$ where $\delta = o(y_0)$. Then, using $p_k = O(n^{2/3-\epsilon})$ and hence $\frac{y_0 p_k}{q_k} = o(1) = o(\delta)$, we have

$$\frac{g(y_0 + \delta)}{g(y_0 + \delta + 1)} = 1 + \frac{\delta}{y_0} + o(\frac{\delta}{y_0}).$$

Moreover, for $\delta > 0$

$$\frac{g(y_0)}{g(y_0 + \delta)} = \prod_{j=0}^{\delta-1} (1 + \frac{j}{y_0} + o(\frac{\delta}{y_0})) \le 1 + \frac{\delta(\delta - 1)}{2y_0} + o(\frac{\delta(\delta - 1)}{2y_0}),$$

$$\frac{g(y_0)}{g(y_0 - \delta)} = \prod_{j=1}^{\delta-1} (1 - \frac{j}{y_0} + o(\frac{\delta}{y_0}))^{-1} \le 1 + \frac{\delta(\delta - 1)}{2y_0} + o(\frac{\delta(\delta - 1)}{2y_0}).$$

Choose $\delta = y_0^{1/2}$, then

$$g(y_0 + y_0^{1/2}), g(y_0 - y_0^{1/2}) \ge (\frac{2}{3} + o(1)) g(y_0).$$

As $\sum_{j=-y_0^{1/2}}^{y_0^{1/2}} g(j) \le 1$ and $y^*$ should be in $[y_0 - y_0^{1/2}, y_0 + y_0^{1/2}]$, we obtain

$$g(y^*) \le \frac{3}{4 y_0^{1/2}} = c_5 \frac{q_k^{1/2}}{p_k},$$

which concludes the proof. $\square$

Now, we go back to consider any $\epsilon \frac{\log n}{\log(r-1-\epsilon)}$ many $k$'s in $(\frac{1}{2}\frac{\log n}{\log(r-1-\epsilon)}, \frac{4}{7}\frac{\log n}{\log(r-1-\epsilon)})$. Based on Lemma C.3, the probability that $A^{(1)}$ and $A^{(2)}$ share the same edge configurations between $Q_k$ and $Q_{k+1}$ for all these $k$'s is bounded by

$$\prod_{k=c_7\frac{\log n}{\log(r-1-\epsilon)}}^{(c_7+\epsilon)\frac{\log n}{\log(r-1-\epsilon)}} \mathbb{P}(C_k) \leq \prod_{k=c_7\frac{\log n}{\log(r-1-\epsilon)}}^{(c_7+\epsilon)\frac{\log n}{\log(r-1-\epsilon)}} c_5\frac{q_k^{1/2}}{p_k} < n^{\epsilon \log c_5}\frac{n^{\frac{\epsilon}{2}\frac{\log n}{\log(r-1-\epsilon)}}}{n^{(c_7+\frac{\epsilon}{2})\epsilon\frac{\log n}{\log(r-1-\epsilon)}}}$$

$$< n^{-\frac{\epsilon^2}{3}\frac{\log n}{\log\log n}} = o(n^{-3/2}).$$

**Step 4:** From step 2, we know that the graphs that do not satisfy $|Q_k| \geq (r-1-\epsilon)^{k-1}$ and $p_k \geq (r-1-\epsilon)|Q_k|$ for $k \in (\frac{\epsilon}{5}\frac{\log n}{\log(r-1)}+1, (\frac{2}{3}-\epsilon)\frac{\log n}{\log(r-1)})$ are only $o(n^{-3/2})$ portion of all graphs generated from the configuration model. From step 3, we know that $A^{(1)}$ and $A^{(2)}$ that satisfy the properties of step 2, with probability at most $o(n^{-3/2})$, their subtrees rooted at node $u$ are the same even with DE-1's. Even if all these graphs belong to simple regular graphs, as step 1 tells the portion of simple regular graphs among the graphs generated from the configuration model is $\Omega(n^{-1/2})$, we arrive at the final conclusion that if $A^{(1)}$ and $A^{(2)}$ are sampled from all simple $r$-regular graphs, with probability at least $1 - o(n^{-3/2})/\Omega(n^{-1/2}) = 1 - o(n^{-1})$, a proper DE-GNN can distinguish $(u, A^{(1)})$ and $(u, A^{(2)})$. $\qquad\square$

## D   Discussion on node sets over irregular graphs

Theorem 3.3 focuses on node sets embedded in regular graphs. A natural question therefore arises: how about the power of DEs to distinguish non-isomorphic node sets embedded in irregular graphs that the 1-WL test (also WLGNN) may not distinguish. To answer this question, note that there is some important connection between irregular graphs and regular graphs under the umbrella of the 1-WL test. Actually, the partition of nodes over irregular graphs according to their representations (colors) stably associated by the 1-WL test has *equitable* property [62]. Basically, suppose that the whole node set $V$ can be partitioned into several parts based on the representations (colors) of nodes, $V = \cup_{i=1}^c V_i$, where for an arbitrary $i$, nodes in $V_i$ share the same representations based on the 1-WL test. Then, the induced subgraph of the nodes in $V_i$ is a regular graph for all $i$'s. What's more, for any $i, j$, the number of nodes in $V_j$ that are neighbors of a certain node in $V_i$ is shared by all the nodes in $V_i$, which again shows certain regularity. Theoretically, if we focus on the regular subgraph defined over each $V_i$, we may further leverage DE defined over this subgraph to further distinguish the nodes in $V_i$. In practice, we do not need to work on each subgraph individually. Mostly, the capability of DE indicated by Theorem 3.3 may still help with breaking such regularity.

## E   Proof for The Limitation of DE-1 — Theorem 3.7

We restate Theorem 3.7: Consider any two nodes $v, u \in V$. Consider two tuples $\mathcal{T}_1 = (v, \mathbf{A}^{(1)})$ and $\mathcal{T}_2 = (u, \mathbf{A}^{(2)})$ with graph topologies $A^{(1)}$ and $A^{(2)}$ that correspond to two connected DRGs with a same intersection array. Then, DE-GNN-1 must depend on discriminatory node/edge attributes to distinguish $\mathcal{T}_1$ and $\mathcal{T}_2$.

*Proof.* We are to prove that if $A^{(1)}$ and $A^{(2)}$ correspond to two DRGs with a same intersection array, then the subtrees rooted at any nodes are all same even if the nodes are labeled with any DE-1's (see illustration of a subtree rooted at a node in Fig. 3). Because if the subtrees are same, the only possibility to differentiate two nodes is based on discriminatory node/edge attributes embedded in these subtrees when DE-GNN processes them from the bottom to the top.

We recall the definition of the intersection array of a connected DRG as the following.

**Definition E.1.** The intersection array of a connected DRG with diameter $\triangle$ is an array of integers $\{b_0, b_1, ..., b_{\triangle-1}; c_1, c_2, ..., c_\triangle\}$ such that for any node pair $(u, v) \in V \times V$ that satisfies $\text{SPD}(v, u) = j$, $b_j$ is the number of nodes $w$ that are neighbors of $v$ and satisfy $\text{SPD}(w, u) = j + 1$, and $c_j$ is the number of nodes $w$ that are neighbors of $v$ and satisfy $\text{SPD}(w, u) = j - 1$.

Figure 4: The subtrees rooted at two nodes in the Shrikhande graph and the $4 \times 4$ rook's graph respectively: In the left two graphs, the black nodes are the target nodes who structural representations are to be learnt. Different colors of the nodes correspond to different DE-1 (SPDs) with respect to the target nodes. For both the Shrikhande graph and the $4 \times 4$ rook's graph, the subtrees rooted at the black nodes share the same spreading colors as shown in the right. As these two graphs are DRGs with the same intersection array, the configuration of colors (DE-1) of children only depends on the color (DE-1) of their father node.

The definition of DRG implies the following lemma.

**Lemma E.2.** Suppose each node is associated with SPD as DE-1. Consider a graph with the intersection array $\{b_0, b_1, ..., b_{\triangle-1}; c_1, c_2, ..., c_\triangle\}$. For an arbitrary node $u$, any node $v$ in the subtree rooted at $u$, with SPD$(v, u) = j$, has children including $b_j$ nodes $w$ with DE-1 satisfying SPD$(w, u) = j + 1$, $c_j$ nodes with DE-1 satisfying SPD$= j - 1$, and $b_0 - b_j - c_j$ nodes $w$ with DE-1 satisfying SPD$(w, u) = j + 1$.

*Proof.* A DRG with the intersection array $\{b_0, b_1, ..., b_{\triangle-1}; c_1, c_2, ..., c_\triangle\}$ is a $b_0$-regular graph and thus $v$ has $b_0$ neighbors. All the neighbors of $v$ become children of $v$ in the subtree. As SPD$(v, u) = j$, we know that SPDs from the neighbors of $v$ to $u$ are in $\{j - 1, j, j + 1\}$. According to the definition of intersection array. we know the numbers of neighbors of $v$ with different SPDs, $j - 1, j, j + 1$, exactly are $c_j, b_0 - c_j - b_j, b_j$ respectively. $\square$

We start from any node $u$ and construct the subtree rooted at $u$. By using the Lemma E.2, it is obvious that subtrees for any nodes from DRGs with a same intersection array are all the same even if all the nodes use SPD as DE-1. An illustrative example of this result is shown in Fig. 4.

The next step is to generalize SPD as DE-1 to the list of landing probability as DE-1. Actually, the following lemma indicates that in DRGs, SPD and the list of landing probability are bijective.

**Lemma E.3.** In any connected DRG with the same intersection array, the number of walks of given length between nodes depends only on the SPD between these vertices.

*Proof.* Given any node $u \in V$ over the DRG with the intersection array $\mathcal{L} = \{b_0, b_1, ..., b_{\triangle-1}; c_1, c_2, ..., c_\triangle\}$, denote the number of walks of length $l$ from $u$ to another node $v$ is $f(v, l)$. We need to prove that $f(v, l)$ can be written as a function $g(\text{SPD}(u, v), l, \mathcal{L})$, which is indepdent from the node identities. We prove this result via induction over the length of walks denoted by $l$. The statement is trivial for $l = 1$. Suppose the statement is true for $l = l_0 - 1$, we consider the case $l = l_0$.

Because of the definition of walks, there is recurrence relation

$$f(u, v, l_0) = \sum_{w \in \mathcal{N}_v} f(u, w, l_0 - 1)$$
$$= \sum_{w:\text{SPD}(w,u)=\text{SPD}(v,u)-1} f(u, w, l_0 - 1) + \sum_{w:\text{SPD}(w,u)=\text{SPD}(v,u)+1} f(u, w, l_0 - 1)$$

Because the assumption of induction, we have

$$f(u, v, l_0) = c_{\text{SPD}(v,u)} g(\text{SPD}(v, u) - 1, l_0 - 1, \mathcal{L}) + b_{\text{SPD}(v,u)} g(\text{SPD}(v, u) + 1, l_0 - 1, \mathcal{L})$$

which only depends on SPD$(v, u), l_0$ and $\mathcal{L}$ and thus can be written as $g(\text{SPD}(u, v), l_0, \mathcal{L})$. $\square$

Figure 5: $K$-hop aggregation does necessarily better the discriminatory power. Consider using DE-GNN-1 and DEA-GNN-1-2-hop to learn the structural representation of the nodes colored by black. We choose SPD as DE-1. Both models require at least two layers to distinguish two black nodes. DEA-GNN-1-2-hop cannot decrease the number of layers by a factor of 2.

Actually, Lemma E.3 is closely related the argument in [63], which claims the same result for walks over one DRG. However, Lemma E.3 extends the argument to any DRGs with the same intersection array. As DRGs are $b_0$-regular graphs, there is a bijective mapping between the list of landing probabilities and the list of walks of different length. Therefore, this is a bijective mapping between SPD and the list of landing probabilities, which concludes the proof. □

## F   Proof for DEA-GNN— Corollary 3.8 and Further Discussion

DEA-GNN contains a general aggregation procedure assisted by DE-1 (Eq. (4)). As we discussed in Section 3.3, as DEA-GNN allows to use DEs as extra node features as DE-GNN in the same time, it has at least the same representation power as DE-GNN. Therefore, Theorem 3.3 and Corollary 3.4 are still true for DEA-GNN. So **our first question** is whether DEA-GNN shares the same limitation with DE-GNN with DE-1 for node representation learning over DRGs. Later, we will prove our confirmation to this question.

An interesting case in practice is to set the DE-1 in Eq. (4) as SPD $\zeta(u|v) = \zeta_{spd}(u|v)$. For some $K \geq 1$, we specify the aggregation as

$$\text{AGG}(\{f_2(h_u^{(l)}, \mathbf{A}_{vu})\}_{u \in \mathcal{N}_v}) \to \text{AGG}(\{(f_2(h_u^{(l)}, \mathbf{A}_{vu}), \zeta_{spd}(u|v))\}_{\zeta_{spd}(u|v) \leq K}), \qquad (9)$$

which means that the aggregation happens among $K$-hop neighbors. If the model is to learn the representation of a node subset with size $p$, We term this model as DEA-GNN-$p$-$K$-hop. **The second question** is to investigate whether DEA-GNN-$p$-K-hop may decrease the number of layers of DE-GNN required in Theorem 3.3 and Corollary 3.4 by a factor of $K$. This result is not trivial. For example, Fig. 5 shows two trees whose root nodes are the target nodes to learn structural representation. Obviously, DE-GNN-1 needs two layers to distinguish these two root nodes. However, DEA-GNN-1-2-hop may not decrease the number of layers by a factor of 2 and thus still needs two layers. This is because the set aggregation in Eq. (9) may decrease the discriminatory power of those features interacted with graph structures. However, next, we can still prove the confirmation to this second questions.

**Confirmation to the first question.** We formally restate the conclusion to prove: Consider any two nodes $w_1, w_2 \in V$. Consider two tuples $\mathcal{T}_1 = (w_1, A^{(1)})$ and $\mathcal{T}_2 = (w_2, A^{(2)})$ with graph topologies $A^{(1)}$ and $A^{(2)}$ that correspond to two connected DRGs with a same intersection array. Then, DEA-GNN-1 must depend on discriminatory node/edge attributes to distinguish $\mathcal{T}_1$ and $\mathcal{T}_2$.

*Proof.* Because Lemma E.3 implies that in all DRGs with the same intersection array, there is a bijective mapping between SPD and the list of landing probabilities. Therefore, we only need to focus on the case that uses SPDs as DE-1, which both appears as extra node attributes and features in aggregation (see Eq. (4)). As the statement is about the limitation, we consider the most expressive case, *i.e.*, the aggregation appearing among every pair of nodes.

Recall the tree structure to compute DE-GNN is termed as the subtree rooted at some node. Now, we call the tree structure to compute DEA-GNN for the structural representation of a node $w$ as the *extended* subtree rooted at $w$, which has the same utility as the tree structure for DE-GNN (Fig. 3).

Suppose we are to learn the structural representation of $w$. We are going to prove by induction that with arbitrary number of layers, if no discriminatory node/edge attributes are available, any node $v$ will be associated with an representation vector that only depends on SPD$(v|w)$ and the intersection array of the graph. If this is true, then we know that any nodes in DRGs with a same intersection array will have the same representation output by DEA-GNN with an arbitrary number of layers.

Recall that the number of layers is $L$. Obviously, when $L = 0$, the node representations only depend on SPD$(v|w)$ and thus satisfy the statement. Suppose the statement is true when $L = L_0 - 1$, consider the case when $L = L_0$. Denote the representation of a node $v$ after $L_0 - 1$ layers as $h_v^{(L_0-1)}$. Then, its representation after $L_0$ layers is

$$h_v^{(L_0)} = f_1(h_v^{(L_0-1)}, \text{AGG}\{(f_2(h_u^{(L_0-1)}), \zeta_{spd}(u|v))\}_{u \in V}).$$

Note that we omit the edge attributes $\mathbf{A}_{uv}$ due to the requirement of the statement. Consider another node $v'$ who satisfies SPD$(v'|w)$ = SPD$(v|w)$. For this, we only need to prove that the following two components are the same and thus $h_v^{(L_0)} = h_{v'}^{(L_0)}$:

$$h_v^{(L_0-1)} = h_{v'}^{(L_0-1)}, \tag{10}$$

$$\{(h_u^{(L_0-1)}, \zeta_{spd}(u|v))\}_{u \in V} = \{(h_u^{(L_0-1)}, \zeta_{spd}(u|v'))\}_{u \in V}. \tag{11}$$

The Eq. (10) is directly due to the assumption of induction. To prove Eq. (11), we first partition all nodes in $V$ according to $\zeta_{spd}(u|v), \zeta_{spd}(u|v')$ by defining $S_v(a) = \{u \in V | \zeta_{spd}(u|v) = a\}$ and $S_{v'}(a) = \{u \in V | \zeta_{spd}(u|v') = a\}$. We further partition these two sets according to $\zeta_{spd}(u|w), \zeta_{spd}(u|w)$ by defining $S_v(a, b) = \{u \in S_v(a) | \zeta_{spd}(u|w) = b\}$ and $S_{v'}(a, b) = \{u \in S_{v'}(a) | \zeta_{spd}(u|w) = b\}$. By using the definition of DRG, we know that $|S_v(a, b)| = |S_{v'}(a, b)|$ and such cardinality only depends on the intersection array $\mathcal{L}$. Moreover, using the assumption of induction, all nodes u in $S_v(a, b), S_{v'}(a, b)$ share the same representation $h_u^{(L_0-1)}$. Combining the fact that $|S_v(a, b)| = |S_{v'}(a, b)|$ and the fact the nodes in these two sets hold the same representation, we may claim the second Eq. (11) is true, which concludes the proof.

$\square$

**Confirmation to the second question.** We formally restate the conclusion to prove: Given two fixed-sized sets $S^{(1)}, S^{(2)} \subset V$, $|S^{(1)}| = |S^{(2)}| = p$. Consider two tuples $\mathcal{T}^{(1)} = (S^{(1)}, \mathbf{A}^{(1)})$ and $\mathcal{T}^{(2)} = (S^{(2)}, \mathbf{A}^{(2)})$ in the most difficult setting when features $\mathbf{A}^{(1)}$ and $\mathbf{A}^{(2)}$ are only different in graph topologies specified by $A^{(1)}$ and $A^{(2)}$ respectively. Suppose $A^{(1)}$ and $A^{(2)}$ are uniformly independently sampled from all r-regular graphs over $V$ where $3 \leq r < (2 \log n)^{1/2}$. Then, for some constant $\epsilon > 0$ and constant positive integer $K$, there exist a proper DEA-GNN-p-K-hop with layers $L < \lceil (\frac{1}{2} + \epsilon) \frac{\log n}{K \log(r-1)} \rceil$, using DE-p $\zeta(u|S^{(1)})$, $\zeta(u|S^{(2)})$ for all $u \in V$ such that with probability $1 - o(n^{-1})$, its outputs $\Gamma(\mathcal{T}^{(1)}) \neq \Gamma(\mathcal{T}^{(2)})$.

*Proof.* Similar to the proof of Theorem 3.3, we use Lemma C.1 and focus on the case $S^{(1)} = S^{(2)} = \{w\}$ and the initial node attribute for each node $u$ are $\zeta_{spd}(u|w)$. Most of the logic of the proof is the same as that of Theorem 3.3. We only need to take care of the step 3 of the proof of Theorem 3.3, as in DEA-GNN-1-K-hop, a node $v$ with SDP$(v|w) = k$ will aggregation representations of nodes $u$ even with SDP$(u|w) \in [k - K, k + K]$. Therefore, we need to redefine the edge configuration in the step 3 of the proof of Theorem 3.3.

Recall $Q_k = \{v \in V | \text{SDP}(v|w) = k\}$. We define the edge configuration between $Q_k$ and $\cup_{i \in [k+1, k+K]} Q_i$ as a list $\bar{C}_k = ((a_{1,k,1}, a_{2,k,1}, ...), (a_{1,k,2}, a_{2,k,2}, ...), ..., (a_{1,k,K}, a_{2,k,K}, ...))$, where $a_{m,k,j}$ is the number of nodes in $Q_{k+j}$ of which each connects to exactly $m$ nodes in $Q_{k+j-1}$. Note that two different $\bar{C}_k$'s will lead to two different representations of $(w, A^{(1)})$ and $(w, A^{(2)})$ after layers $\lceil \frac{k}{K} \rceil + 2$ as DEA-GNN-1-K-hop uses at most 2 layers yield $\{h_u^{(2)} | \zeta_{spd}(u|w, A^{(1)}) = k\} \neq \{h_u^{(2)} | \zeta_{spd}(u|w, A^{(2)}) = k\}$ and uses at most $\lceil \frac{k}{K} \rceil$ layers to propagate such difference to the target node $w$.

Actually, this definition of edge configuration is nothing but a concatenation of the edge configurations $\{C_i\}_{i \in [k, k+K-1]}$ where $C_i$ is the edge configuration between $Q_i$ and $Q_{i+1}$ as defined in the proof of Theorem 3.3. Then, we use the statement of step 3 on $C_i$ to characterize the probabilitic property of

$\bar{C}_k$. For each $k \in (\frac{1}{2}\frac{\log n}{\log(r-1-\epsilon)}, \frac{4}{7}\frac{\log n}{\log(r-1-\epsilon)})$, each type of edge configuration $\bar{C}_k$ appears with only limited probability $\mathbb{P}(\bar{C}_k) = \Pi_{i=k}^{k+K-1}\mathbb{P}(\bar{C}_i) = \Pi_{i=k}^{k+K-1}O(\frac{n^{1/2}}{p_i})$. Then, we consider $\epsilon\frac{\log n}{K\log(r-1-\epsilon)}$ many $k$'s in $(\frac{1}{2}\frac{\log n}{\log(r-1-\epsilon)}, \frac{4}{7}\frac{\log n}{\log(r-1-\epsilon)})$ such that these $k$'s hold the same integral interval $K$ and can be denoted as $k_0, k_0+K, ....$ The probability that $A^{(1)}$ and $A^{(2)}$ have all the same edge configurations for these $k$'s is bounded by

$$\Pi_{k\in\{k_0, k_0+K,...\}}\mathbb{P}(\bar{C}_k) = \Pi_{i=k_0}^{k_0+\epsilon\frac{\log n}{\log(r-1-\epsilon)}-1}\mathbb{P}(C_i) \sim o(n^{-\frac{3}{2}}).$$

Therefore, we only need to consider edge configurations for $k \in (\frac{1}{2}\frac{\log n}{\log(r-1-\epsilon)}, (\frac{1}{2}+\epsilon)\frac{\log n}{\log(r-1-\epsilon)})$ to distinguish $A^{(1)}$ and $A^{(2)}$. And within $\lceil(\frac{1}{2}+\epsilon)\frac{\log n}{K\log(r-1-\epsilon)}\rceil + 2$, DEA-GNN-1-$K$-hop yields different representaions for $(w, A^{(1)})$ and $(w, A^{(2)})$. Note that the constant 2 may be merged in $\epsilon$ for simplicity, which concludes the proof. □

# G  Details of the Experiments

## G.1  Datasets

The three air traffic networks for Task 1, Brazil-Airports, Europe-Airports, and USA-Airports were collected by [64] from the government websites throughout the year 2016 and were used to evaluate algorithms to learn structural representations of nodes [5, 6]. Networks are built such that nodes represent airports and there exists an edge between two nodes if there are commercial flights between them. Brazil-Airports is a network with 131 nodes, 1,038 edges and diameter 5; Europe-Airports is a network with 399 nodes, 5,995 edges and also diameter 5; USA-Airports is a network with 1,190 nodes, 13,599 edges and diameter 8. In each dataset, the airports are divided into 4 different levels according to the annual passengers flow distribution by 3 quantiles: 25%, 50%, 75%. The goal is to infer the level of an airport using solely the connectivity pattern of them.

Tasks 2 & 3 were carried out on three other datasets used by SEAL [9] to facilitate comparison study: C.ele, NS and PB. C.ele [65] is a neural network of C. elegans with 297 nodes, 2,148 edges and 3241 triangles (closed node triads), and diameter of 5, in which nodes are neurons and edges are neural linkage between them. NS [66] is a network of collaboration relationship between scientists specialized in network science, comprising of 1461 nodes 2742 edges and 3764 triangles. PB [64] is a network of reference relationships between political post web-pages, consisting of 1222 nodes, 16714 edges, and of diameter 8. Following [9, 10], for Task 2 & 3, we remove all links or triangles in testing sets from graph structure during the training phase to avoid label leakage.

## G.2  Baseline Details

We have five baselines based on GNNs and one baseline, struc2vec [5], based on kernels using handcrafted structural features. We first introduce the implementation of struc2vec and then discuss other baselines.

*Struc2vec* is implemented in a 2-phase manner. In the first phase, embeddings for all the nodes are learned by running the n-gram framework over a constructed graph based on structural similarity kernels. We directly use the code provided by the original paper [5] [4]. In the second phase, the embeddings of nodes that in the target node set are concatenated and further fed into an one-layer fully connected neural network to make further inference.

Regarding other GNN-based baselines, *GCN* is implemented according to Equation (9) of [20] with self-loops added. *SAGE* is implemented according to Algorithm 1 of Section 3.1 in [21]. Mean pooling is used as the neighborhood aggregation function. *GIN* is implemented by adapting the code provided by the original paper [16] [5], where we use the sum-pooling aggregation and multi-linear perception to aggregate neighbors. In all three baselines described above, ReLU nonlinearities are applied to the output of each hidden layer, followed by a Dropout layer. *PGNN* layer is implemented by adapting the code provided by the original paper [10] [6]. *SEAL* is implemented by adapting the code

provided by the original paper [7]. As we focus on learning structural representation with inductive capability, all the five GNN-based methods use node degrees as input features if node attributes are not available.

Final readout layers are tailored to suit different tasks. For Task 1 since the task is node classification, the final layer for all baselines is a one-layer neural network followed by a cross entropy loss. Tasks 2 & 3 have slightly more complex readout layers since the target entity for prediction is a node set of size 2 or 3. Note that SEAL is specifically designed for Task 2 and has its own readout that uses SortPooling over all node representations over the ego-networks of node-pairs [23]. we refer the readers to the original paper for details [9]. For all the other baselines, to make a fair comparison, we use the following difference-pooling: Suppose the target node set is $S$ and the representation of node $v$ for $v \in S$ is denoted by $h_v$, then we readout the representation of $S$ as

$$z = \sum_{u,v \in S} |h_v - h_u| \tag{12}$$

where $|\cdot|$ denotes component-wise absolute value. Note that Tasks 2 & 3 are to predict the existence of a link / triangle. So we use the inner product $\langle w, z \rangle$ where $w \in \mathbb{R}^d$ is a trainable final projection vector, and feed this product into the binary cross entropy loss to train the models.

### G.3 DE-GNN Variants Details.

**Minibatch training based on ego-network extraction.** To understand our detailed framework it is helpful to first discuss the minibatch training of GCN, although the original GCN is trained in a full-batch manner [20]. To train GCN in minibatchs, we first extract, for each target node $v$ in a given minibatch, an ego-network centering at $v$ within $L$-hop neighbors by doing a depth-$L$ BFS from $v$, denoted by $G_v$. Here, $L$ is the number of GCN layers intended to be used in the model. Note that the representation of node $v$ via using $L$-layer GCN over $G_v$ is the same as that via using $L$-layer GCN over the whole graph. If node attributes are not available for GCN or other WLGNNs, we may use the degree of each node as its node attributes.

Our models are implemented by following the above mini-batch training framework. For a target node set $S$, we first extract the union of ego-networks centering at any nodes in $S$ within $L$-hop neighbors. We call the union of ego-networks as the ego-network around $S$, denoted by $G_S = \cup_{v \in S} G_v$. Note that even if $S$ has multiple nodes, $G_S$ can be extracted as a whole by running BFS. All the edges of $G_S$ between nodes that are both in $S$ will be further removed, which is denoted by $G'_S$. For $G'_S$, we associate each node $u$ in these ego-networks with the DE $\zeta(u|S)$ as extra node attributes. Specifically, we use a simple aggregation for Eq. (1):

$$\zeta(u|S) = \frac{1}{|S|} \sum_{v \in S} \zeta(u|v) \tag{13}$$

Next, we detail the different versions of $\zeta(u|v)$ used by different variants of DE-GNN.

**DE-GNN-SPD.** This variant sets $\zeta(u|v)$, $v \in S$ and $u \in G_S$ as a one-hot vector of the truncated shortest-path-distance between $u$ and $v$. That is,

$$\zeta_{spd}(u|v) = \text{one hot}(\min(\text{SPD}(u, v), d_{max})), \tag{14}$$

where $d_{max}$ is the maximum distance to be encoded. As a result, the $\zeta_{spd}(u|v)$ is a vector of length $d_{\max} + 1$. The pairwise SPDs can either be pre-computed in preprocessing stage, or be computed by traversing the extracted ego-networks on the fly. The $d_{max}(\leq L)$ helps prevent overfitting the noise in an overly large neighborhood.

**Compare DE-GNN-SPD with SEAL.** DE-GNN-SPD, when used for link prediction, is similar to SEAL [9] in a sense that we both encode the distance between any node and the two nodes in the target node-pairs. However, we are fundamentally different from SEAL as SEAL uses graph-level readout of all nodes in the ego-networks. SEAL also has no discussion on the expressive power of distance encoding. Their intention of node labeling as they reported is just to let the model know which node-pair in the extracted ego-networks is the target node-pair. Moreover, the specific DE $\zeta(u|S)$ for a node-pair $S = \{v_1, v_2\}$ used in SEAL is a one-hot encoding of the value

$1 + \min(\text{SPD}(u, v_1), \text{SPD}(u, v_2)) + (d/2)[(d/2) + (d\%2) - 1]$, where $d = \text{SPD}(u, v_1) + \text{SPD}(u, v_2)$. The dimension of this DE is $O(d_{\max}^2)$ which is higher than our DE (Eq. (14)) used in DE-GNN-SPD, which may result in model overfitting for large $d_{\max}$

**DE-GNN-LP.** This variant sets $\zeta(u|v)$, $v \in S$ and $u \in G_S$ as landing probabilities of random walks (of different lengths) from node $v$ to node $u$:

$$\zeta_{lp}(u|v) = ((W^{(0)})_{vu}, (W^{(1)})_{vu}, (W^{(2)})_{vu}, ..., (W^{(d_{\text{rw}})})_{vu}) \tag{15}$$

where $W^{(k)} = (AD^{-1})^k$ is the $k$-step random walk matrix, $d_{\text{rw}}$ is the max step number of random walks. Notice that in principle, $\zeta_{lp}(u|v)$ encodes the distance information at a finer granularity than $\zeta_{spd}(u|S)$. This is because $\text{SPD}(u, v)$ can be inferred from $\zeta_{lp}(u|v)$ as the index of the first non-zero random walk feature. However, in practice we observe that such encoding does not always bring a significant performance gain.

For both DE-GNN-SPD and DE-GNN-LP, we use the same 1-hop neighborhood aggregation as GCN. DE can also be used to control the message passing as shown in Eq. (4). We further discuss two variants by incorporating DE-GNN-SPD and Eq. (4).

**DEA-GNN-SPD.** DEA-GNN-SPD is built on top of DE-GNN-SPD, using the same extra node features $\zeta_{spd}(u|S)$, but allows for multi-hop aggregation by specifying Eq. (4) as

$$\text{AGG}(\{f_2(h_u^{(l)}, \mathbf{A}_{vu})\}_{u \in \mathcal{N}_v}) \rightarrow \text{AGG}(\{(f_2(h_u^{(l)}, \mathbf{A}_{vu}), \zeta_{spd}(u|v))\}_{\zeta_{spd}(u|v) \leq K}).$$

In experiments, we choose $K = 2, 3$, which means that each node aggregates representations of other nodes that are not only its direct neighbors but also its (exclusive) 2-hop and even 3-hop neighbors. As we do not have edge attributes in our data, we omit $\mathbf{A}_{vu}$. Our implementation of the aggregation for the layer $l$ follows

$$h_v^{(l+1)} = \sum_{k=1}^{K} \text{Relu}\left(\frac{1}{|S_{v,k}| + 1}\left(h_v^{(l)} + \sum_{u \in S_{v,k}} h_u^{(l)} \Theta^{(lk)}\right)\right), \ S_{v,k} = \{u|\zeta_{spd}(u|v) = k\}.$$

where $\Theta^{(lk)}$ is a trainable weight matrix and for each $k$, we aggregate $k$-hop neighbors via a GCN layer with a self-loop. Note that when implementing DEA-GNN-SPD, we need to extract the ego-network of nodes within $LK$-hops, if DEA-GNN-SPD has $L$ layers.

**DEA-GNN-PR.** DEA-GNN-SPD is also built on top of DE-GNN-SPD, but the propagation is by specifying Eq. (4) as

$$\text{AGG}(\{f_2(h_u^{(l)}, \mathbf{A}_{vu})\}_{u \in \mathcal{N}_v}) \rightarrow \text{AGG}(\{(f_2(h_u^{(l)}, \mathbf{A}_{vu}), \zeta_{gpr}(u|v))\}_{v \in V}).$$

As the aggregation is over the whole node set, this model does not extract the ego-networks but uses the entire graphs. For the layer $l$, we further specify the above aggregation by using

$$h_v^{(l+1)} = \text{Relu}\left(\sum_{u \in V} \zeta_{ppr}(u|v) h_u^{(l)} \Theta^{(l)}\right)$$

where $\Theta^{(l)}$ is a trainable weight matrix and $\zeta_{ppr}(u|v)$ is a specific form of $\zeta_{gpr}(u|v)$ based on Personalized Pagerank scores [67], i.e,

$$\zeta_{ppr}(u|v) = [\sum_{k=0}^{\infty}(0.9W)^k]_{uv} = [(I - 0.9W)^{-1}]_{uv}.$$

Note that the above $0.9$ is a hyper-parameter. As we are just willing to show the use case, $0.9$ is set as a heuristic and is not obtained via parameter tuning. Other values may yield better performance. Other types of PageRank scores may be used, *e.g.*, heat-kernel PageRank scores [68], time-dependent PageRank scores [69].

We compare DEA-GNN-PR with all other methods, which yields the following Table 2. DEA-GNN-PR performs worse than DEA-GNN-SPD while it still works much better than WLGNNs in link and triangle predictions. Comparing these observations with the statements on GDC [32], we argue that missing DEs as node attributes is the key that limits the performance of link prediction via GDC.

| Data / Method | Nodes (Task 1): Average Accuracy | | | Node-pairs (Task 2): AUC | | | Node-triads (Task 3): AUC | |
|---|---|---|---|---|---|---|---|---|
| | Bra.-Airports | Eur.-Airports | USA-Airports | C.elegans | NS | PB | C.elegans | NS |
| GCN [20] | 64.55±4.18 | 54.83±2.69 | 56.58±1.11 | 74.03±0.99 | 74.21±1.72 | 89.78±0.99 | 80.94±0.51 | 81.72±1.50 |
| SAGE [21] | 70.65±5.33 | 56.29±3.21 | 50.85±2.83 | 73.91±0.32 | 79.96±1.44 | 90.23±0.74 | 84.72±0.40 | 84.06±1.14 |
| GIN [16] | 71.89±3.60† | 57.05±4.08 | 58.87±2.12 | 75.58±0.59 | 87.75±0.56 | 91.11±0.52 | 86.42±1.12† | 94.59±0.66† |
| Struc2vec [5] | 70.88±4.26 | 57.94±4.01† | 61.92±2.61† | 72.11±0.31 | 82.76±0.59 | 90.47±0.60 | 77.72±0.58 | 81.93±0.61 |
| PGNN [10] | N/A | N/A | N/A | 78.20±0.33 | 94.88±0.77 | 89.72±0.32 | 86.36±0.74 | 79.36±1.49 |
| SEAL [9] | N/A | N/A | N/A | 88.26±0.56† | 98.55±0.32† | 94.18±0.57† | N/A | N/A |
| DE-GNN-SPD | **73.28±2.47** | 56.98±2.79 | **63.10±0.68*** | **89.37±0.17*** | **99.09±0.79** | **94.95±0.37*** | **92.17±0.72*** | **99.65±0.40*** |
| DE-GNN-LP | **75.10±3.80*** | 58.41±3.20* | **64.16±1.70*** | 86.27±0.33 | 98.01±0.55 | 91.45±0.41 | 86.24±0.18 | **99.31±0.12*** |
| DEA-GNN-SPD | **75.37±3.25*** | 57.99±2.39* | **63.28±1.59** | **90.05±0.26*** | **99.43±0.63*** | **94.49±0.24*** | **93.35±0.65*** | **99.84±0.14*** |
| DEA-GNN-PR | **73.26±4.08** | 51.41±2.39 | 50.34±1.50 | 83.07±0.77 | **99.46±0.37*** | 92.68±0.57 | 83.15±1.11 | **99.86±0.03*** |

Table 2: Model performance (including DEA-GNN-PR) in Average Accuracy and Area Under the ROC Curve (AUC) (mean in percentage ± 95% confidence level). † highlights the best baselines. *, **bold font**, **bold font*** respectively highlights the case where our proposed model's performance: exceeds the best baseline on average, exceeds by 70% confidence, exceeds by 95% confidence.

## G.4  Model performance without validation datasets

We have confirmed with the original authors of Struc2vec [5] and SEAL [9] that the performance of these two baselines reported in their papers do not use validation set. The performance therein is the best testing results ever achieved when models are being trained till convergence. We think that it is necessary to include validation datasets to achieve fair comparison and therefore reported the results with validation datasets in the main text. We put the results without validation datasets here. Under both experimental settings, we draw similar conclusions for our models in comparison with the baselines.

| Data / Method | Nodes (Task 1): Average Accuracy | | | Node-pairs (Task 2): AUC | | | Node-triads (Task 3): AUC | |
|---|---|---|---|---|---|---|---|---|
| | Bra.-airports | Eur.-airports | USA-airports | C.elegans | NS | PB | C.elegans | NS |
| GCN [20] | 82.01±3.09 | 55.56±2.90 | 60.08±4.20 | 78.10±1.24 | 81.92±0.90 | 90.04±1.23 | 83.15±0.50 | 88.56±3.54 |
| GraphSAGE [21] | 81.48±6.26 | 63.41±5.87 | 54.62±4.29 | 77.20±1.26 | 86.16±2.95 | 90.73±1.38 | 86.84±1.13 | 87.98±4.58 |
| GIN [16] | 85.19±5.85† | 65.47±2.99† | 63.45±5.29 | 79.79±1.42 | 89.84±3.62 | 91.47±0.58 | 89.37±0.44† | 95.48±0.64† |
| stuc2vec [5] | 84.08±9.30 | 65.30±5.60 | 68.09±2.50† | 74.17±0.37 | 87.43±2.48 | 90.96±0.26 | 83.66±3.26 | 87.53±3.12 |
| PGNN [10] | N/A | N/A | N/A | 80.76±0.98 | 94.99±1.44 | 90.21±0.78 | 87.18±0.52 | 83.39±1.74 |
| SEAL [9] | N/A | N/A | N/A | 90.30±1.35† | 98.85±0.47† | 94.72±0.46† | N/A | N/A |
| DE-GNN-SPD | **87.78±3.95** | 65.62±3.27* | **70.95±0.80*** | **90.67±0.91*** | **99.50±0.87** | **95.21±0.53** | **92.54±0.63*** | **99.91±0.13*** |
| DE-GNN-LP | **88.36±4.62** | 66.25±2.07* | 69.33±1.33* | 88.48±0.46 | 98.24±0.67 | 92.43±1.29 | 87.65±0.71 | **99.74±0.21*** |
| DEA-GNN-SPD | **88.43±3.24*** | 67.14±1.57* | **71.07±0.63*** | **91.09±0.96*** | **99.52±0.88** | **95.15±0.24*** | **93.28±0.51*** | **99.94±0.06*** |
| DEA-GNN-PR | **87.30±2.52** | 64.38±1.78 | 62.46±1.18 | 84.23±0.55 | **99.58±0.46*** | 92.94±0.76 | 84.61±0.58 | **99.95±0.06*** |

Table 3: Model performance without validation set (including DEA-GNN-PR) in Average Accuracy and Area Under the ROC Curve (AUC) (mean in percentage ± 95% confidence level). † highlights the best baselines. *, **bold font**, **bold font*** respectively highlights the case where our proposed model's performance: exceeds the best baseline on average, exceeds by 70% confidence, exceeds by 95% confidence.

We report additional results of our model on Task 2 and 3 versus other baselines, measured by average accuracy without validation set in Table 4. Similar observations as reported in the main text can be drawn from both Table 3 and 4: the strongest baselines are given by SEAL [9] and GIN [16], while our DEA-GNN variants further significantly outperform those baselines on all tasks.

| Data / Method | Node-pairs (Task 2) Average Accuracy | | | Node-triads (Task 3) Average Accuracy | |
|---|---|---|---|---|---|
| | C.elegans | NS | PB | C.elegans | NS |
| GCN [20] | 65.01± 0.45 | 74.69± 1.39 | 78.35± 1.69 | 69.47± 2.69 | 82.16± 1.27 |
| SAGE [21] | 67.96± 0.90 | 78.29± 2.40 | 83.53± 1.41 | 76.67± 0.72 | 88.45± 0.65 |
| GIN [16] | 69.45±1.24 | 82.17±1.07 | 83.01±0.83 | 77.60±0.85† | 92.67±1.44† |
| Struc2vec [5] | 68.41±2.57 | 80.64±0.92 | 74.43±3.48 | 71.06±3.86 | 83.42±1.67 |
| PGNN [10] | 71.40±1.68 | 91.04±0.85 | 86.44±1.14 | 76.34±0.23 | 80.67±0.49 |
| SEAL [9] | 83.90±0.97† | 97.89±0.40† | 88.92±0.95† | N/A | N/A |
| DE-GNN-SPD | 83.78±0.88 | **99.54±0.28*** | **89.82±0.82*** | **90.82±0.62*** | **100.0±0.00*** |
| DE-GNN-LP | 70.96±1.27 | 96.96±0.57 | 84.37±1.03 | **82.90±0.69*** | **100.0±0.00*** |
| DEA-GNN-SPD | **84.81±0.69*** | **99.77±0.24*** | **89.83±0.98*** | **91.24±0.22*** | **100.0±0.00*** |
| DEA-GNN-PR | 76.95±0.85 | **99.88±0.19*** | 85.82±0.87 | **83.18±0.71*** | **100.0±0.00*** |

Table 4: Model performance without validation set (including DEA-GNN-PR) in Average Accuracy (mean in percentage ± 95% confidence level). † highlights the best baselines. *, **bold font**, **bold font*** respectively highlights the case where our proposed model's performance: exceeds the best baseline on average, exceeds by 70% confidence, exceeds by 95% confidence.

## G.5  Hyperparameters Tuning.

Table 5 lists the most important hyperparameters' at a glance, which applies to both the baselines and our proposed models. Grid search is used to find the best hyperparameters combination. The models

are sufficiently trained till the cross entropy loss converges and we report the best model by running each for 20 times over different random seed. For more details please refer to the code attached.

| Hyperparameters | Value / Range | Notes |
|---|---|---|
| batch size | 64, 128 | |
| learning rate | 1e-4 | |
| optimizer | SGD | stochastic gradient descent |
| conv. layers | 1, 2, 3 | struc2vec does not follow this setting |
| conv. hidden dim. | 20, 50, 80, 100 | struc2vec does not follow this setting |
| dropout | 0, 0.2 | |
| $d_{rw}$ | 3, 4 | #steps of random walk, valid only for DE-GNN-RW |
| $d_{spd}$ | 3, 4 | maximum shortest path distance for DE-GNN-SPD variants |
| prop_depth | 1, 2, 3 | the number of hops of message in one layer, only valid for DEA-GNN-SPD, 1 for all the others |

Table 5: List of hyperparameters and their value / range.

# H  A Brief Introduction of Higher-Order Weisfeiler-Lehman Tests

We mentioned higher-order WL tests in our context, especially on the 2-WL test. There are different definitions of the $k$-WL test ($k \geq 2$), while in this work we followed the definition in [39]. Note that the $k$-WL test here also corresponds to the $k$-WL' test in [40] and the $k$-FWL test in [27], and are equivalent to the $k + 1$-WL tests in [26, 27].

**The $k$-WL test** ($k \geq 2$) follows the following coloring procedure:

1. For each $k$-tuple of node set $V_i = (v_{i_1}, v_{i_2}, ..., v_{i_k}) \in V^k$, $i \in [n^k]$, we initialize $V_i$ with a color denoted by $C_i^{(0)}$. These colors satisfies that for two $k$-tuples, $V_i$ and $V_j$, $C_i^{(0)}$ and $C_j^{(0)}$ are the same if and only if for $a, b \in [k]$ (1) $v_{i_a} = v_{i_b} \Leftrightarrow v_{j_a} = v_{j_b}$ and (2) $(v_{i_a}, v_{i_b}) \in E \Leftrightarrow (v_{i'_a}, v_{i'_b}) \in E$.

2. For each $k$-tuple $V_i$ and $u \in V$, define $N(V_i; u)$ as a sequence of $k$-tuples such that $N(V_i; u) = ((u, v_{i_2}, ..., v_{i_k}), (v_{i_1}, u, ..., v_{i_k}), (v_{i_1}, v_{i_2}, ..., u))$. Then, the color of $V_i$ can be updated via the following mapping:

$$C_i^{(l+1)} \leftarrow g(C_i^{(l)}, \{(C_j^{(l)} | V_j \in N(V_i; u))\}_{u \in V}),$$

where $g(\cdot)$ is injective coloring.

3. For each step $l$, $\{C_i^{(l)}\}_{i \in [n^k]}$ is a coloring configuration of the graph $G$, which is essentially a multi-set. If two graphs have different coloring configurations, these two graphs are determined to be non-isomorphic, while the inverse is not true.

Note that the step 2 essentially requires to aggregate colors of $nk$-tuples and thus even when $k = 2$, the 2-WL test may not leverage the sparsity of graph structure to keep good scalability. Ring-GNN [30] and PPGN [27] essentially try to achieve the expressive power of the 2-WL test. They are not scalable to process large graphs and they were only evaluated for entire-graph-level tasks such as graph classification and graph regression [27, 30].