[Reviews · NeurIPS 2020]

Review 1

Summary and Contributions: [edit: see "additional feedback" section for response to author rebuttal] This paper considers the problem of learning representations of specific subgraphs of a larger graph, in such a way that subgraphs of non-isomorphic graphs should be assigned different representations. Specifically, they note that most GNNs can only distinguish things that the 1-WL test can distinguish, and propose a new method of computing graph features (called "distance encoding") to get around this restriction. The distance encoding the authors propose appears to work as follows. Suppose we want to compute a representation of a subgraph S of a graph G. Instead of running a GNN on G and then pooling across the subgraph, the subgraph S is used to modify the features of all of the nodes in G, in particular based on the distances between those nodes in G and the nodes in S (this is DEGNN). Optionally, the subgraph is also used to modify the edge structure by, conceptually, adding additional edges and/or edge features based on pairwise distances in G (this is DEAGNN). Then, the GNN is run as normal on this graph. The authors include theoretical results for regular graphs and distance-regular graphs, showing that simple GNNs cannot distinguish either of these, and that their DEGNN models can distinguish regular graphs with high probability, but cannot distinguish distance-regular graphs. They also include experiments on node feature prediction, link prediction, and triangle (clique) prediction, and show strong results. Although their approach seems to work in practice and has some theoretical justifications, I think the paper focuses too much on the specific feature encoding they use, while obscuring the fact that they are considering a different, simpler problem, by assuming access to a particular target subgraph in the input that the 1-WL test and other GNN methods do not have. This simpler problem is interesting, but not well described, and I think the paper could be greatly improved by revising it to focus on this general idea and not on the specific encoding.

Strengths: The proposed distance encoding is similar to prior work, but seems to generalize that prior work, and seems like an interesting starting point for more sophisticated feature encodings. The paper has a good balance of theoretical and experimental results. Theoretically, they demonstrate sets of graphs that their method can distinguish but simpler GNNs fail to distinguish, as well as sets of graphs for which their method cannot distinguish. Empirically, they do a thorough evaluation of their model on node, edge, and triangle prediction, and show strong results versus baselines.

Weaknesses: Although admittedly I am not an expert in graph isomorphism tests and similar literature, it seems to me that the additional "power" of the DEGNN method is not due to the particular encoding proposed here, but instead comes from focusing on an easier problem. The authors motivate their approach by discussing the Weisfeiler-Lehhman tests, which can be used to decide that particular pairs of graphs are not isomorphic. The idea of WL tests is to take each graph, compute a representation of that graph, and then compare if the representations are equal; if they are not, the graphs must not be isomorphic. Graph neural networks can be viewed as doing something similar, in that they compute representations of graphs based on local features, and it can be shown that for the graph isomorphism task, simple GNNs have equivalent power to the 1-WL test. However, the task the authors consider here is simpler: given two graphs, and also a particular subset S of nodes within each graph, determine if there is an isomorphism between the two graphs that also restricts to an isomorphism between those node subsets (by computing a representation of each graph that depends also on S, and seeing if those are equal). As a special case, if the subset is of size one, this is asking whether there is an isomorphism between the two graphs such that the chosen nodes map to each other under the isomorphism. This task is easier, since it has significantly reduced the set of matchings that we need to consider. The authors then make use of this additional information by encoding the query subgraph S as a collection of distances of other nodes to nodes in that query set. But if you are trying to compute a representation of a specific subgraph S, there are much simpler things you could do! For instance, just adding a separate node feature to each node that says whether it is in S, and then running a GNN on that. I think that this would have equivalent representational power without adding any of the complexity of the distance encoding, since a GNN could learn to compute distance encodings on its own when given the starting set of nodes S. (This kind of conditioning is not novel, and in fact when the set S is known in advance many GNN applications do something like this, see for instance section 3.1 of "Gated Graph Sequence Neural Networks".) Corollary 3.4 in the paper does show that you can use a single-subgraph structural representation to compute a full-graph structural representation, by doing injective pooling across the structural representations for all nodes. But it is important to note that this is much more computationally expensive than doing a 1-WL test or a normal GNN pass, since to compute the representation for each individual node, you must run a separate GNN with different node features. And, again, the representative power of this does not seem to depend on the distance encoding, but simply on the fact that the GNN is allowed to condition on a particular node set. I do think that characterizing the representative power of pooled small-node-set structural representations is an interesting thing to do. But in my opinion this paper does not make a clear distinction between (a) using an input query subset to affect node representations, and (b) using the specific feature encoding they propose based on random walks. I think it would be much more compelling to focus the theoretical aspect of the paper on the difference in power between GNNs that compute graph representations (without having access to a subgraph S) and GNNs that compute small-node-set representations (which are allowed to condition on a particular subgraph S as an extra feature), and then present the distance encoding as a particular special case of this general idea. Additionally, the approach is motivated based on graph isomorphism tests, but none of the experiments have to do with graph isomorphisms! It seems important to actually evaluate the proposed method on a graph isomorphism problem and demonstrate that the model can actually learn that kind of task in practice. Notably missing is any discussion of the actual readout operation and loss function used for the experiments with their models. The appendix includes a few details about readout layers for baselines, but nothing for the DEGNN models, and even this is not particularly easy to understand.

Correctness: I found a few potential issues with the definitions and claims made in the paper: In definition 3.1, there seems to be a mismatch between the name "distance encoding" and the extremely broad definition. For instance, under this definition, a constant function is a distance encoding, as is a function that counts the degree of the current node, or that counts the number of unique paths between the node and S, or that computes eigenvalues of subgraphs of the adjacency matrix, or many similar things. Is that intentional? Theorem 3.3 is written in terms of "some small constant ε > 0". This is an extremely imprecise statement. How small is small? There is no upper bound. Could it grow like n? Faster? Slower? I have not read the proof in detail, but I skimmed it and could not find any definition of ε whatsoever. Does Theorem 3.7 depend on the particular use of distance encoding? Again, I only skimmed the proof, but it seems to rely on the specific case of shortest path distance, even though the theorem statement is more general.

Clarity: On a high level, I think that the paper does not put enough emphasis on how their task differs from the standard formulation, and in particular the difference between substructure representations of SPECIFIC substructures, vs whole graph representations. Many prior GNN approaches can be seen as producing either a single graph representation, or simultaneously producing representations for individual nodes or node pairs. This paper seems to be different because it instead is focused on producing a representation for a SINGLE set of nodes, without having to compute representations for every node at once. This is, in my opinion, the core difference between their approach and the GNN approaches similar to the 1-WL test, but this is not clearly stated; it took me until section 3.1 to realize that a single, specific subset S of interest is provided at input time.

Relation to Prior Work: I am not very familiar with the prior work in this space, so I cannot speak strongly to this. However, the authors do seem to cite a wide range of previous contributions.

Reproducibility: No

Additional Feedback: Line 45: "review them in details" should be "review them in detail" Line 243: "to do the following change" is unclear. Change to what? Is it figure 1b? Also, it would be better to actually write out the full model definition instead of just writing down a substitution. Line 296: "has recently attracted much significance to" I don't understand what this means. Broader impact section: My understanding is that this section is supposed to focus on ethical impacts. However, much of this section is in fact a summary of the rest of the paper, as a replacement for a conclusion, which seems contrary to the intended purpose. Also, on line 385: choosing datasets using "rules that are irrelevant to ethics" is NOT a sufficient way to ensure ethical research! Ignoring ethical issues does not make them go away, instead it is important to incorporate ethics into the decision making process. (That said, I don't think that caused any particular problems in this case.) ===== Edit after author response ===== I had not considered that the node set could be the entire graph. It seems that in that case the probability is 1-o(1), which could increase to 1 arbitrarily slowly, but I suppose this is still more powerful than the WL test in this setting, which would remain small. I'm still struggling to wrap my head around how this paper relates to prior work and known results. My current understanding after reading the rebuttal, which may be incomplete: - The 1WL test cannot distinguish certain graphs, but does not use a specific node set. - "WLGNN" models also cannot distinguish those graphs, and also do not use a specific node set. - Methods like GraphSAGE and SEAL do use a specific node set, so they are already more powerful than the 1WL test. But the power of these has not been theoretically established. - Distance encodings are a way of making features that DO use a specific node set. - The "everything in the set gets 1" simple idea I mentioned in my initial review is technically a distance encoding. It also can be more powerful than the 1WL test, because it uses the node set. But if the node set is the whole graph, it isn't any more powerful than the 1WL test. - On the other hand, the distance encoding based on SPD is still more powerful than the 1WL test, even when there is NOT a specific node set, according to the claims in the rebuttal. (This is somewhat surprising to me!) - Some prior work has used distance encoding before, but they have not proved any bounds on how powerful they are. I think that the abstract and introduction of the paper could be improved to clarify the above relationships and position this work appropriately with respect to the existing literature. In particular it seems important to emphasize that: - giving access to the node set is already more powerful than the WL test, and so prior approaches that have access to the node set aren't necessarily limited to the WL test (even though they may not be as powerful as distance encodings) - the primary contribution of this work is the formalization and mathematical characterization of the power of distance encodings, not the novelty of the encoding itself (as the authors note in the "Novelty" section of the rebuttal) It seems to me that the n=k case (and probably the small-k case as well) strongly depends (through Lemma C.1) on the aggregation function being injective. How do you ensure that this is the case? If this is a heuristic choice, what do you choose (since the example of sum pooling seems clearly not injective)? If this is a learned aggregation process, is it still provable that DEGNN can distinguish nodes using this learned aggregation instead of an ideal one? It would be good to address this in the next revision. I still do not understand what value epsilon should have. Does the statement hold for every epsilon > 0 that is independent of n? If so, it would be better to say "for any epsilon". Otherwise it seems important to clarify what value epsilon has in the proof, and how small it is. I also still feel that Definition 3.1 is too broad for the name "distance encoding", as it allows many things that are not distances to be called distance encodings. If this generality is desired, I think it would be better to call it something like "target-subset encoding", with distance encodings as a (provably powerful) special case of that. If not, perhaps the definition should be made more specific so that things that are clearly not distances aren't included. (See my initial review for examples.) I am increasing my review to 5 (Marginally below acceptance threshold) in light of the clarifications in the rebuttal.


Review 2

Summary and Contributions: The paper introduces a data augmentation framework for the improvement of the expressiveness of graph neural networks based on the structural information that a graph entity has. Although there are other works that suggest efficient ways of improving the performance of different graph learning algorithms, this work provides an extensive study on how the encoded structural information can be applied on different scenarios either as extra node features or as controller in the aggregation step of a message passing scheme. Moreover, the authors suggest a categorization of the message passing neural networks between the WLGNN, which capitalizes on the computational power of the WL isomorphism test and the DEGNN, as they call their method, which encapsulates the graph-related data augmentation inside already known models.

Strengths: The idea seems to be novel even though there are other works that suggest the utilization of structural information in the node attribute space [1]. The authors provide a unified framework of incorporating different levels of structural dependencies through the parameterization of the \zeta distance encoding function. Moreover, as it is shown in the experimental section, the method seems to outperform some strong baselines in all the examined tasks. Finally, as it is stated in the "Broader Impact" section, the framework can be easily modified and applied to any kind of graph-related scenario. [1] "Are Powerful Graph Neural Nets Necessary? A Dissection on Graph Classification". Ting Chen, Song Bian, Yizhou Sun. arXiv 2019.

Weaknesses: - The authors do not refer to the complexity of their method. In section 3, they claim that "this simple design can be efficiently implemented", however they do not provide any information on the computational cost of Equation (2). In particular, the utilization of powers of random-walk matrices seems to exhibit high complexity, while the special cases of SPD and generalized PageRank scores are, also, expensive. - In Table 1, the authors compare their method against a few baselines. However, particularly for the case of the role prediction, a few stronger competitors have been suggested before, such as the RolX [2] algorithm, which also, incorporates structural features, extracted in an automatic manner and GraphWave [3], which exploits diffusion wavelets to provide encoded information about the graph structure and seems to be one of the strongest structural representation learning baseline. The authors should compare the proposed approach against these competitors as well, in order to have a more complete overview of their contribution. - As it is described in Appendix G1 Section, the employed benchmark datasets are very small ( # nodes < 1500 ), so a 10% test split would provide ~= 150 test samples, not allowing for a safe conclusion of the actual accuracy improvement of Table 1. Moreover, regarding the structural representation tasks, it is natural to assume larger size graphs, where structurally similar nodes could lie in distant positions of the same graph. Thus, it would be nice if the authors utilize larger graphs in the experimentats. [2] "RolX: Structural Role Extraction & Mining in Large Graphs". Keith Henderson et al. In KDD 2012. [3] "GraphWave: Learning Structural Node Embeddings". Claire Donnat, Marinka Zitnik, David Hallac, and Jure Leskovec. In KDD 2018.

Correctness: The method and the empirical methodology seem to be correct.

Clarity: The paper lacks of clarity in many parts. Specifically, in section 3, where the authors introduce their main contribution, the paragraphs are not well-structured and the statements are vague both contextually and syntactically.

Relation to Prior Work: It is clear how the proposed approach differs from previous methods.

Reproducibility: Yes

Additional Feedback: ========Edit After Author's Response=========== I would like to thank the authors for addressing a number of my concerns, however, the response did not change my opinion. In my view, this is a decent paper, but I still think that the paper lacks clarity about a number of technical details and I would like to see more experimental results (even on synthetic datasets to verify the theoretical results).


Review 3

Summary and Contributions: This paper proposes Distance Encoding, a simple yet effective way to enhance the representation power of GNNs by augmenting node features or controlling message passing via pairwise distance. Using node distance to augment GNNs breaks the symmetry among nodes, while without losing the inductive ability of GNNs as compared to those approaches using node identifiers. The paper theoretically shows that almost all r-regular graphs can be distinguished by Distance Encoding where traditional 1-WL and GNNs fail. It also gives a limitation result saying Distance Encoding cannot handle distance regular graphs. The empirical results on node/link/triangle prediction demonstrate better results than baselines.

Strengths: The theoretical analysis is sound and convincing. The method proposed is simple yet effective. The paper is very relevant to the NeurIPS community.

Weaknesses: The theoretical analysis only shows Distance Encoding's better representation power than 1-WL on regular graphs. Although regular graphs are one major type of counter-examples for 1-WL, it is beneficial to discuss or analyze Distance Encoding's power for general graphs. The empirical evaluation on node classification does not use standard benchmark datasets. It is reasonable that Distance Encoding improves over GNNs on feature-less graphs. Is it also useful for graphs with rich features?

Correctness: The claims and method look correct to me, though I didn't check the full proofs. The empirical methodology is in a fair setting.

Clarity: The paper is generally well written, though it can be improved by introducing the core idea earlier. The main idea of Distance Encoding doesn't become clear until page 4. Too many preliminaries precede the main topic. Theorems 2.6 and 2.7 also look a bit straightforward (they both directly generalize theorems in existing works such as GIN), while taking a lot of space.

Relation to Prior Work: Yes. The paper discusses its relations to 1-WL, WLGNN, SEAL, PGNN, etc. very well.

Reproducibility: Yes

Additional Feedback: Can Distance Encoding also enhance 2-WL and higher-order GNNs? ------------------------------------------ After rebuttal: I would like the authors to know that the reviewers and meta-reviewer discussed this paper extensively on its strengths, weaknesses, and areas of improvements. We finally reached a concensus that the paper indeed makes excellent technical contributions while the writing should be improved a lot. My suggestions on writing are: 1) Discuss the differences between DE and node subset conditioning (raised by R1). Explicitly point out DE incorporates the annotation trick and is more powerful than node conditioning (the author response did a good job). Point out that node conditioning is also more powerful than WL (i.e., DE > node conditioning > 1WL). 2) Clarify the contribution is on characterizing DE mathematically, not proposing DE. This can be done by putting the discussions of prior work using DE (such as SEAL and PGNN) from Section 4 to Introduction.

[Author Response · NeurIPS 2020]

We thank the reviewers for their time and valuable feedback for improving the manuscript. All the reviewers agree that our work provides a novel and general technique to improve GNNs in both theory and practice. Reviewers make several valuable suggestion which we are going to address in the final version of the paper. Specifically, **R1** provides a detailed review that we greatly appreciate and raises concerns on our problem definition and the novelty. **R2 and R3** provide actionable insights on enhancing the experiments and generalizing our theory. We particularly thank **R3** for precisely capturing our problem definition and appreciating the theory. Regarding further generalizing our theory, we refer to some initial attempts described in the Sec.D (supplement). Next, we will focus on the rest questions and suggestions.

**\*Problem definition. R1** makes an insightful observation that by fixing two node sets from two graphs, the graph isomorphism testing becomes a simpler problem. However, the conclusion as it relates to our work is wrong. In contrast, our formalization is in fact *more general and more interesting* as it gives additional flexibility—the target node set could be of arbitrary size, even being the entire node set. By sightly revising the union bound in Lem C.1, Thm 3.3 can be directly generalized to the case with arbitrary size $|S^{(1)}| = |S^{(2)}| = k$ and the success probability becomes $1 - o(k/n)$. When $k = n$, the problem reduces to the standard graph isomorphism problem without any node-set information but we still successfully beat WLGNNs/1-WL test with high probability.

Although our theory is rather general, just to restrict the potentially too wide scope of this work, as stated in Sec. 2, we leave the study of entire graph representations for the future and focus on learning the representations of node sets with small $k$ based on their contextual structures, which itself is a big, if not bigger, use case of GNNs. Such a problem formulation seems to be made by us, but actually not: Many previous works on node classification [3-6], link prediction [8-10] held the same concept but we are the first one to unify them, provide rigorous definition, and more importantly derive relevant theory and better algorithms. Methodologically, many GNN baselines such as GraphSAGE[21], SEAL[9], also need the access to the information of the queried node sets to perform ego-network sampling, but all of them insufficiently heuristically use such information. Therefore, we see it as an important contribution to systematically study the benefit of such information (Thm 3.3) and how to mostly use it (via DE).

**\*Novelty.** As **R1, R2** and our Sec. 4 pointed out, previous works proposed some similar idea as DE. However, as claimed, we are the first one to *mathemtically characterize* the power of DE, which governs many empirically successful GNN models (Sec. 4). The importance of theory has already been demonstrated when we try to evaluate **R1**'s trick to annotate the target node set $S$: With a certain choice of $f_3$ (Eq. (2)), DE actually reduces to this trick as a special case. However, the annotation trick may lose some representation power, as it does not give an injective $f_3$ (see its requirement in Def. 3.2): For the case when $S$ is the entire node set ($k = n, S = V$), such a trick obviously fails. For the case when $S$ is small ($k < n, S \subset V$), we may prove that GNN requires much larger $L$ (Thm 3.3) to learn the distance information, and even more architectural effort to achieve the same representation power.

**\*Technical correctness and clarity. R1** had questions on Thms 3.3 and 3.7. $\epsilon$ is independent from $n$ and thus we use "constant". Thm 3.7 generalizes short path distances to other DEs via Lem. E.3. We feel sorry to cause this ambiguity.

**\*Complexity. R1** questioned the complexity of our model for the whole graph representation (Cor 3.4). Again, please note that the whole graph representation is not our primary focus. For rebuttal, first, DEGNN-n (viewing the entire node set $V$ as the target $S$) follows the same complexity as WLGNN but with more representation power while one needs to pre-compute distance encoding for every two nodes (with complexity $O(|E|d)$ where $d$ is the graph diameter). Second, we agree that DEGNN-1 (viewing each node as a separate $S$ and then aggregating their representations) for the whole graph representation is more complex than WLGNN. However, RingGNN [29] and PPGN [30], two most recently proposed provably more powerful GNNs, are even more complex, as they require 3-mode tensors. DEGNN-1 can use 3-mode tensors to compute all node representations simultaneously and allows utilizing sparse graph structures to propagate features. However, RingGNN (PPGN), if using sparse graph structures will break their theoretical foundation. **R2** questioned the complexity of DE computation. Actually, in practice, we only need to compute short path distances or random walks within constant hops (3-4 hops) whose complexity is constant times $|E|$, which is the same as that of feature propagation in GNN. In our experiments, the computation of DE costs less than 5% of the cpu time.

**\*Experiments.** We respectfully disagree with **R2**'s doubt on the safety of our evaluation. Regarding the dataset size, we have larger datasets for the link prediction task with 16k+ links. For node classification, we did not intentionally choose small graphs but used the same benchmarks as Struc2vec[5]. Moreover, we demonstrated the stable advantage of our models via providing the 95% significance scores (20-time tests). We further follow **R2**'s suggestion to compare with two more baselines RolX(RX)[3] and GraphWave(GW)[6] over three Airports datasets and obtain their results (Brazil 62.86(RX), 73.07(GW); Europe 54.50(RX), 56.56(GW); USA 52.52(RX), 54.55(GW)) which are far worse than DE-GNN. We further follow **R2 and R3**'s suggestions to evaluate our model over *an even larger graph (7k+ nodes) with richer features*, Actor (recently in GEOM-GCN (Pei et al, ICLR'20)) and get the results:

| GCN | SAGE | GIN | Struc2vec | GEOM-GCN | DE-GNN-SP | DEA-GNN |
|---|---|---|---|---|---|---|
| 29.9±1.0 | 35.4±0.4† | 27.4±0.7 | 30.04±0.82 | 31.82±0.91 | **36.40±0.75**\* | **36.32±0.37**\* |

DE(A)-GNN here propagates DEs to gather node structural representations and concatenates them with node features to predict node labels. GCN/GIN work bad due to the graph's heterophily. The results justify DEs' further significance to larger graphs, especially heterophilic ones. All the results are obtained under the same setting as in our paper.

In the final version, we will clarify our problem definition, algorithmic complexity and merge the above experiments.

[Meta-Review · NeurIPS 2020]

This exciting paper introduces some interesting and novel theoretical contributions to the graph neural network literature. The authors also verified some of their theoretical findings empirically as well. This paper is worth presenting at NeurIPS with the condition that the authors will address the concerns raised by the reviewers on writing and clarity. This paper has valuable contributions in better characterizing 1-WL's power, yet it steps too large directly to using distance while skipping the discussion of those somewhat more straightforward conditioning ways (such as annotation, etc.) It seems like there is a too big jump from 1-WL directly to DE without discussing how much you gain by just doing more straightforward conditioning. We would suggest the authors address this in the camera-ready version of the paper by revising the writing a bit. More concretely after discussion with other reviewers and in particular with reviewer 3, we have identified that the following changes needs to be done for the camera ready version of the paper: 1. Discuss the differences between DE and node subset conditioning. Explicitly point out distance encoding incorporates the annotation trick and is more powerful than node conditioning (the author response did a good job). Point out that node conditioning is also more powerful than WL (i.e., DE > node conditioning > 1WL). 2. Clarify the contribution is on characterizing DE theoretically, not proposing the DE. This can be done by putting some discussions of prior work using DE (such as SEAL and PGNN) in Section 4 or to the introduction.